# Maternal spindle transfer overcomes embryo developmental arrest caused by ooplasmic defects in mice

Nuno Costa-Borges[1]*, Katharina Spath[2,3], Irene Miguel-Escalada[4], Enric Mestres[1], Rosa Balmaseda[5], Anna Serafín[5], Maria Garcia-Jiménez[1], Ivette Vanrell[1], Jesús González[5], Klaus Rink[1], Dagan Wells[2,3], Gloria Calderón[1]

[1]Embryotools, Parc Cientific de Barcelona, Barcelona, Spain; [2]Nuffield Department of Women's and Reproductive Health, University of Oxford, Oxford, United Kingdom; [3]Juno Genetics, Winchester House, Oxford Science Park, Oxford, United Kingdom; [4]Genomics and Bioinformatics, Centre for Genomic Regulation, Barcelona, Spain; [5]PCB Animal Facility, Parc Cientific de Barcelona, Barcelona, Spain

**Abstract** The developmental potential of early embryos is mainly dictated by the quality of the oocyte. Here, we explore the utility of the maternal spindle transfer (MST) technique as a reproductive approach to enhance oocyte developmental competence. Our proof-of-concept experiments show that replacement of the entire cytoplasm of oocytes from a sensitive mouse strain overcomes massive embryo developmental arrest characteristic of non-manipulated oocytes. Genetic analysis confirmed minimal carryover of mtDNA following MST. Resulting mice showed low heteroplasmy levels in multiple organs at adult age, normal histology and fertility. Mice were followed for five generations (F5), revealing that heteroplasmy was reduced in F2 mice and was undetectable in the subsequent generations. This pre-clinical model demonstrates the high efficiency and potential of the MST technique, not only to prevent the transmission of mtDNA mutations, but also as a new potential treatment for patients with certain forms of infertility refractory to current clinical strategies.

*For correspondence:
nuno.borges@embryotools.com

Competing interests: The authors declare that no competing interests exist.

## Introduction

Infertility disorders are a growing problem that affects millions of couples worldwide (*WHO, 2017*). Although assisted reproductive technologies (ARTs) have evolved and can now successfully address many challenging cases (*Huang and Rosenwaks, 2014*; *Niederberger et al., 2018*), conventional IVF treatment continues to fail a significant percentage of infertile women, with many ultimately ending-up being enrolled in egg donation programs (*Lutjen et al., 1984*; *Sauer et al., 1990*; *Trounson et al., 1983*). The use of donated oocytes is effective at significantly improving the chances of successful IVF treatment, however, the resultant children are not genetically related to the intended-mothers. Therefore, it is desirable to develop new reproductive strategies that can allow the treatment of these patients with genetically related oocytes.

Oocyte quality is defined as the competence of the oocyte to develop into a chromosomally normal blastocyst with potential to sustain a pregnancy up to a healthy live birth. Frequently, poor quality oocytes fail to fertilize or produce embryos that arrest during the first stages of development (*Hardy et al., 2001*; *Meskhi and Seif, 2006*; *Pellicer et al., 1995*) either due to nuclear or cytoplasmic defects (*Conti and Franciosi, 2018*; *Eppig, 1996*; *Liu and Keefe, 2004*). Accumulated evidence suggests that aberrant meiosis or early developmental failure is caused mainly by deficiencies in the oocyte cytoplasmic machinery (*Hoffmann et al., 2012*; *Liu et al., 2003*; *Liu et al., 1999*; *Liu et al., 2000*; *Liu and Keefe, 2007*; *Reader et al., 2017*), which contains a vast diversity of critical

**eLife digest** Infertility is a growing problem that affects millions of people worldwide. Medical procedures known as in vitro fertilization (IVF) help many individuals experiencing infertility to have children. Typically in IVF, a woman's egg cells are collected, fertilized with sperm from a chosen male and grown for a few days in a laboratory, before returning them to the woman's body to continue to develop.

However, there are some women whose egg cells cannot develop into a healthy baby after they have been fertilized. Many of these patients use egg cells from donors, instead. This greatly improves the chances of the IVF treatment being successful, but the resultant children are not genetically related to the intended mothers.

Previous studies suggested that a cell compartment known as the cytoplasm plays a crucial role in allowing fertilized egg cells to develop normally. A new technique known as maternal spindle transfer, often shortened to MST, makes it possible to replace the entire cytoplasm of a compromised egg cell. This is achieved by transplanting the genetic material of the compromised egg cell into a donor egg cell with healthier cytoplasm that has previously had its own genetic material removed. Using this technique, it is possible to generate human egg cells for IVF that have the genetic material from the intended mother without the defects in the cytoplasm that may be responsible for infertility. However, it is not clear whether this approach would be a safe and effective way to treat infertility in humans.

Costa-Borges et al. applied MST to infertile female mice and found that the technique could permanently correct deficiencies in the cytoplasms of poor quality egg cells, allowing the mice to give birth to healthy offspring. Further experiments studied the offspring and their descendants over several generations and found that they also had higher quality egg cells and normal levels of fertility.

These findings open up the possibility of developing new treatments for infertility caused by problems with egg cells, so experiments involving human egg cells are now being performed to evaluate the safety and effectiveness of the technique.

components, including organelles, mRNAs, proteins, ribosomes and many other factors (*Bianchi et al., 2015*; *Sathananthan, 1997*). Mitochondria are the most numerous organelles in the cytoplasm and play an essential role by supplying the ATP needed for the oocyte to support critical events, such as: maturation, spindle formation and segregation of chromosomes and chromatids (*Chappel, 2013*; *May-Panloup et al., 2007*). Dysfunctions at the mitochondrial level and deficiencies affecting other cytoplasmic factors have been correlated with inadequate oocyte developmental competence (*Eichenlaub-Ritter, 2012*; *Liu et al., 2002*; *Van Blerkom, 2011*; *Van Blerkom et al., 1995*), particularly in older infertile patients (*Babayev and Seli, 2015*; *Fragouli et al., 2015*; *Igarashi et al., 2016*; *Wells, 2017*).

Techniques like cytoplasmic transfer (*Cohen et al., 1998*; *Lanzendorf et al., 1999*) or the injection of purified mitochondria (*Fakih MHSM et al., 2015*; *Kristensen et al., 2017* have been proposed as potential methods to restore the viability of compromised oocytes in IVF patients with a history of poor embryo development or repeated implantation failures with conventional treatments. Although live births have been reported following the use of these techniques (*Cohen et al., 1998*; *Fakih MHSM et al., 2015*; *Huang et al., 1999*; *Lanzendorf et al., 1999*) their safety and/or benefits to treat infertility has been questioned. Cytoplasmic transfer experiments were abandoned due to concerns that heteroplasmy (i.e., the co-existence of two distinct mtDNA genomes) might have negative clinical consequences (*Darbandi et al., 2017*; *Isasi et al., 2016*; *Kristensen et al., 2017*). An alternative strategy, which avoided heteroplasmy by utilizing autologous injection of mitochondria from the patient's own germline cells attracted much attention as a possible new treatment to revitalize deficient oocytes (*Johnson et al., 2004*; *White et al., 2012*). Multiple studies in animal models showed apparent benefits of the addition of mitochondria to oocytes of compromised quality (*El Shourbagy et al., 2006*; *Hua et al., 2007*; *Yi et al., 2007*) and IVF births were reported after transfer of oogonial precursor cell-derived mitochondria (*Fakih MHSM et al., 2015*). However, the source and quality of the mitochondria used are unclear and a recent randomized clinical study

conducted using mitochondria derived from autologous oogonial stem cells failed to demonstrate improvements in embryo developmental or clinical outcomes (*Labarta et al., 2019*). Thus, current data from human clinical research do not support the notion that the addition of further mitochondria derived from the same individual is capable of correcting cytoplasmic deficiencies (mitochondria or other) that may be present in poor quality oocytes. Furthermore, the safety of the procedure is yet to be verified. Of note, a recent study suggested that autologous mitochondrial supplementation may induce a phenotypic effect in the heart of resultant mice (*St John et al., 2019*).

An approach that may offer greater promise in terms of its capacity to address infertility problems of maternal (oocyte) origin is the transfer of the nuclear genome from an affected oocyte or zygote into a new 'healthy' cytoplasm. These techniques, known globally as mitochondrial replacement techniques (MRTs) were originally proposed to prevent the transmission of inherited mitochondrial diseases (*Craven et al., 2010*; *Hyslop et al., 2016*; *Paull et al., 2013*; *Tachibana et al., 2009*). Indeed, a clinical application of maternal spindle transfer (MST) to prevent the transmission of Leigh Syndrome was recently reported, resulting the birth of an unaffected child (*Zhang et al., 2017*). However, the potential of MRTs to overcome infertility remains unclear, as most studies utilizing this approach have not had this as their main focus, instead concentrating on their potential to avoid mitochondrial diseases; examination of nuclear-cytoplasmic interactions in oocytes and zygotes (*Liu and Keefe, 2004*; *Liu and Keefe, 2007*); the origin of female aneuploidies *Palermo et al., 2002*; or the decreased developmental capability of aged oocytes in animal models (*Yamada and Egli, 2017*).

Here, we explored the feasibility of the MST technique as a reproductive tool to overcome embryo developmental arrest. To test our hypothesis, a detailed series of proof-of-concept experiments were conducted to assess the safety and the efficiency of the technique using mouse models, which, in a clinical context, could represent donors and patients with oocytes of good and poor developmental competence, respectively. Additionally, advanced molecular techniques were used to evaluate in detail the heteroplasmy levels induced by the procedure in early embryonic-stages and in multiple important organs, including some with high metabolic demand, collected from male and female mice generated by MST. The mice were bred and followed up to ascertain their health, fertility and welfare, as well as, to study the fate of the heteroplasmy in the offspring of the MST female progenitors over five generations.

## Results

### MST among sibling B6CBAF1 oocytes is feasible without impairing embryo development

In a first set of experiments we aimed to optimize the MST protocol and to determine whether the manipulation of the spindle-chromosome complex is feasible without impairing the developmental potential of reconstructed oocytes. We performed reciprocal MST among sibling oocytes from the mouse hybrid *B6CBAF1* strain (*Figure 1a*). Enucleation and reconstruction (karyoplast-cytoplast fusion) of oocytes were first assessed with freshly collected oocytes. Enucleation was successful in 98.9% of oocytes (n = 790) and reconstruction was achieved in 96.1% (n = 321), confirmed using a microscope with polarized light that allows visualization of the birefringence of the spindle microtubules (*Figure 1b–c* and *Figure 1—figure supplement 1*). Next, MST was carried out with both fresh and cryopreserved *B6CBAF1* oocytes that were vitrified and warmed using the open *Cryotop* system (97.7% survival, n = 600). In this set of experiments, spindles were taken from fresh oocytes and transferred into either fresh (*fresh-sp/fresh-cyt*) or vitrified-warmed cytoplasts (*fresh-sp/vitrified-cyt*) and vice-versa, that is spindles from vitrified oocytes transferred to fresh (*vitrified-sp/fresh-cyt*) or vitrified-warm cytoplasts (*vitrified-sp/vitrified-cyt*). The resultant oocytes from the different groups were then fixed after reconstruction and processed for evaluation of the spindle apparatus and chromosomes distribution by immunofluorescence microscopy (*Figure 1e–f* and *Figure 1—figure supplement 2*). All oocytes analyzed presented a spindle with a normal barrel shape and with the chromosomes aligned at the MII plate (*fresh-sp/fresh-cyt* n = 20, *fresh-sp/vitrified-cyt* n = 15, *vitrified-sp/fresh-cyt* n = 15, *vitrified-sp/vitrified-cyt* n = 16; *Figure 1e–f*), regardless of whether fresh or vitrified gametes were used as spindle or cytoplast donors (*Figure 1—figure supplement 2*). These observations indicated that the conditions used to perform the manipulation of the spindle-

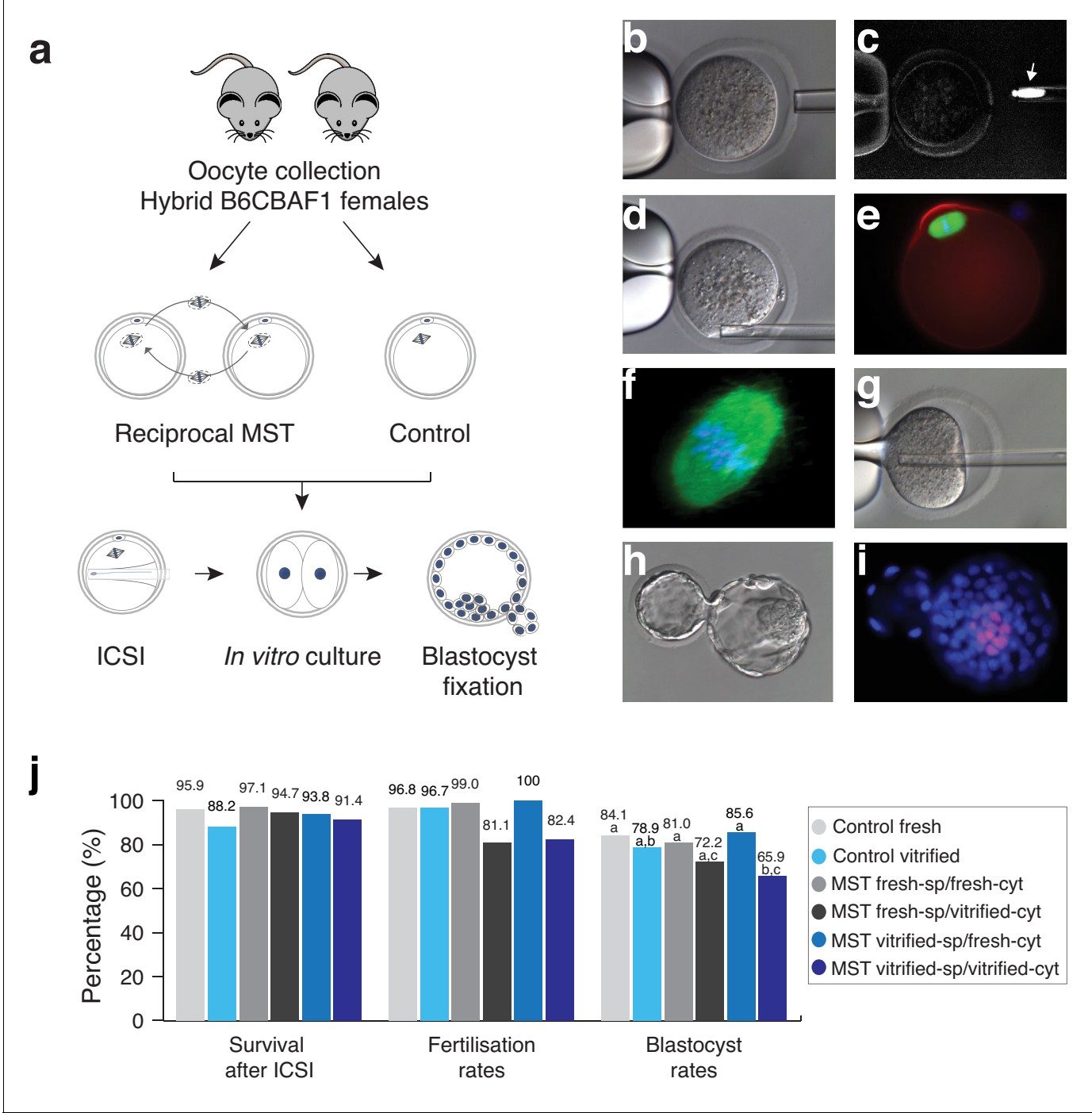

**Figure 1.** Maternal spindle transfer (MST) between sibling fresh B6CBAF1 mouse oocytes does not impair embryo development. (**a**) Schematic representation of the experimental design used to validate the different steps of the technique. (**b**) Detail of the enucleation procedure with confirmation of the spindle isolation under polarized light. (**c**) The birefringence of the meiotic spindle is indicated by an arrow. (**d**) Details of oocyte reconstruction by placing the spindle transfer in the perivitelline space of the enucleated oocyte. (**e**) Representative oocyte reconstructed by MST and processed by immunofluorescence for detection of microtubules (green), microfilaments (red) and DNA (blue). (**f**) Confocal microscopy detail of the meiotic spindle structure in an oocyte reconstructed by MST at a high magnification (600x) showing a normal barrel shape spindle (green) and aligned chromosomes in the metaphase plate (blue). (**g**) Piezo- ICSI performed with a blunt-end pipette in a MST oocyte. (**h**) Hatching blastocyst generated by MST at 120 hr post-ICSI. (**i**) Fixed MST blastocyst processed for total cell counts. (**j**) ICSI survival, fertilisation and blastocyst rates in sibling fresh and vitrified oocytes processed by MST and non-manipulated controls. See also *Figure 1—figure supplements 1* and *2*.

The online version of this article includes the following figure supplement(s) for figure 1:

*Figure 1 continued on next page*

*Figure 1 continued*

**Figure supplement 1.** Representative images of the MST protocol.

**Figure supplement 2.** Representative immunofluorescence images of B6CBAF1 strain fresh (left) or vitrified (right) oocytes that have been non-manipulated (control) or used as spindle (sp) donors for MST procedures.

chromosome complex were neither damaging to its structure nor altering of the distribution of the chromosomes. Furthermore, there was no evidence that the procedure was inducing premature activation of the oocytes.

Subsequently, in an independent set of samples, we compared the in vitro development of reciprocal MST experiments using fresh and vitrified *B6CBAF1* oocytes, after insemination by ICSI (*Figure 1j* and *Figure 1—figure supplement 1*). High enucleation (98.7%, n = 399) and fusion (98.2%, n = 394) rates were achieved in all MST groups (see also *Table 1*) and almost all oocytes that were prepared with fresh (99%, n = 100) or vitrified (100%, n = 90) spindles, and transferred into fresh cytoplasts, developed to the two-cell stage on the next morning (*Figure 1h–j* and *Table 1*). Interestingly, a significantly lower proportion of inseminated oocytes composed of vitrified spindles transferred into vitrified cytoplasts (*vitrified-sp/vitrified-cyt*) developed to the two-cell stage (82.4%, n = 85) compared with non-manipulated fresh (96.8%, n = 94, p=0.001) or vitrified (96.7%, n = 90, p=0.001) controls. Poorer development was also observed for the *fresh-sp/vitrified-cyt* group (81.1%, n = 90) (*Figure 1j* and *Table 1*). On the contrary, when spindles from vitrified oocytes were transferred into fresh cytoplasts (*vitrified-sp/fresh-cyt*, n = 90), two-cell stage (100%) and blastocyst formation (85.6%) rates were high and equivalent to fresh controls (96.8% and 84.1%, respectively) or to MST oocytes where fresh spindles were transferred into fresh cytoplasts (*fresh-st/fresh-cyt*, n = 100, 99% and 81%, respectively) (*Figure 1j* and *Table 1*). Additionally, the mean number of total cells (mean ± SD, n) in the blastocysts obtained in the *fresh-st/fresh-cyt* group (177.8 ± 26.7, n = 81) was equivalent to controls (192 ± 29.5; n = 79). No differences were found either in the number of inner cell mass cells that were positive for the *Oct4* pluripotency marker between *fresh-st/fresh-cyt* and control groups (22.4 ± 3.5; n = 14 *versus* 25.3 ± 5.6; n = 10, see also *Figure 1i*). Taken together, the experiments performed among sibling *B6CBAF1* oocytes, showed that MST is technically feasible in the mouse without impacting the in vitro developmental competence of the oocyte. Experiments indicate that vitrification induces changes that make cryopreserved oocytes unsuitable for use as cytoplasts. However, the spindle apparatus does not appear to be damaged during vitrification or MST procedures. When recipient cytoplasts were derived from fresh oocytes, blastocyst development rates were equivalent to those obtained for non-manipulated controls, regardless of whether the spindle originated from a fresh or vitrified oocyte.

## MST overcomes embryo development arrest in NZB oocytes

After careful optimization and validation of the different steps of the MST protocol, the effectiveness of the technique as a strategy to overcome embryo developmental arrest was evaluated. Two

**Table 1.** Efficiency and in vitro developmental rates of B6CBAF1 mouse oocytes processed by MST using fresh and vitrified oocytes.

| | n oocytes processed by MST | | | | In vitro development for up 96 hr post-ICSI | | | | |
|---|---|---|---|---|---|---|---|---|---|
| Group | n initial | Enucleated (%) | Fused (%) | ICSI survival (%) | n cultured | Two-cells (%) | Blastocysts (%) | Total cell counts (± SD) | Oct4+ |
| Control fresh | 98 | N/A | N/A | 94 (95.9) | 94 | 91 (96.8)* | 79 (84.1)* | 192.1 (29.5) | 25.3 (5.6) |
| Control vitrified | 102 | N/A | N/A | 90 (88.2) | 90 | 87 (96.7)* | 71 (78.9)*,† | N/A | N/A |
| MST *FreshSp/FreshCyt* | 107 | 107 (100) | 103 (96.2) | 100 (97.1) | 100 | 99 (99)* | 81 (81)* | 177.8 (26.7) | 22.4 (3.5) |
| MST *FreshSp/VitriCyt* | 96 | 96 (100) | 95 (98.9) | 90 (94.7) | 90 | 73 (81.1)† | 65 (72.2)*,‡ | N/A | N/A |
| MST *VitriSp/FreshCyt* | 98 | 96 (97.9) | 96 (100) | 90 (93.8) | 90 | 90 (100)* | 77 (85.6)* | N/A | N/A |
| MST *VitriSp/VitriCyt* | 98 | 95 (96.9) | 93 (97.9) | 85 (91.4) | 85 | 70 (82.4)† | 56 (65.9)†,‡ | N/A | N/A |

*, †, ‡ Values with different superscripts differ significantly within the same column (p<0.05; Chi-square test or Fisher's test).

different oocyte strains were employed: the hybrid B6CBF1 (resultant from the cross between C57BL/6JRj females and CBA/Jrj males), and the *New Zealand Black* (*NZB/OlaHsd*) strains. The *NZB* strain holds two interesting characteristics. Firstly, *NZB* mice present a poor reproductive performance (*Fernandes et al., 1973*; *Hansen CT and Whitney, 1973*) and, secondly, the genetic background of the *NZB* strain has diverged genetically from most other mouse laboratory strains, including the hybrid *B6CBAF1* strain, accompanied by characteristic differences in mtDNA sequences (*Bielschowsky and Goodall, 1970*). These two features are particularly relevant to the experimental design of this study as, in a clinical context, the *NZB* strain could be considered analogous to a subfertile patient (especially those with a history of poor in vitro embryo development), and the *B6CBAF1* strain, a donor of proven fertility. Additionally, single nucleotide polymorphisms in the divergent mtDNA of the *NZB* strain provides an opportunity to evaluate the carryover of organelles and resultant heteroplasmy induced by MST procedures (see Materials and methods). Experiments were thus carried out between the two mouse strains, so that meiotic spindles were transferred from fresh *NZB* oocytes into fresh *B6CBAF1* cytoplasts and vice-versa (*Figure 2a*). Once reconstructed, oocytes were inseminated using ICSI in parallel with non-manipulated oocytes from both strains and cultured in vitro until the blastocyst stage (*Figure 2a*). Enucleation and fusion rates were identical in both MST groups, and no differences were found in terms of survival to ICSI compared to controls (*Figure 2b* and *Table 2*). As expected, *NZB* control oocytes presented significantly lower fertilization rates than *B6CBAF1* control oocytes, measured as two-cell stage development (*Figure 2b* and *Table 2*). Additionally, while blastocyst formation rates were close to 80% in the *B6CBAF1* control group (77.8%, n = 144), most of the injected oocytes from the *NZB* control group arrested their development before reaching this stage (5.6% developed into blastocysts, n = 159, *Figure 2b* and *Table 2*). Remarkably, when the meiotic spindles from *NZB* oocytes were transferred into *B6CBAF1* cytoplasts (*NZB-sp/B6-cyt*), the blastocyst formation rates were 10-fold higher (51.4%, n = 212, p<0.0001) compared to the non-manipulated NZB control (*Figure 2b* and *Table 2*). In the reciprocal MST group, *B6CBAF1* spindles transferred into *NZB* cytoplasts (*B6-sp/NZB-cyt*), blastocysts were not obtained (0%, n = 110), indicating that cytoplasmic factors are likely to be responsible for the

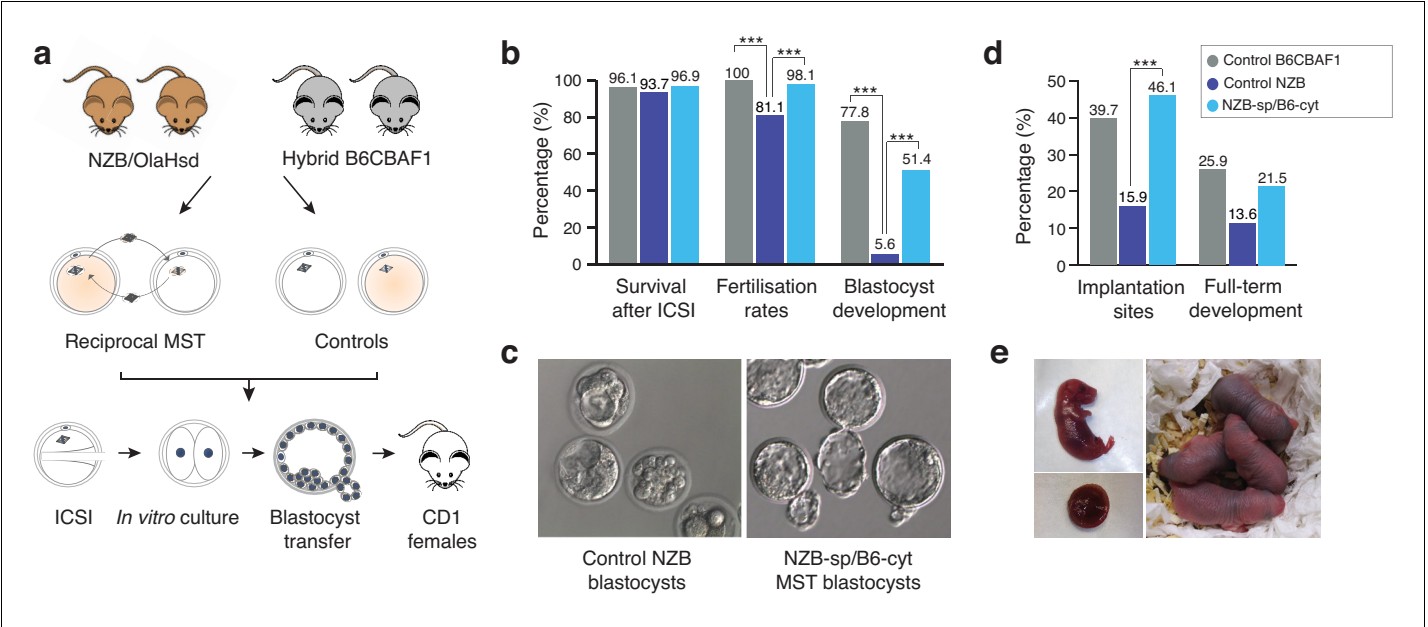

**Figure 2.** Meiotic spindle transfer between NZB/OlaHsd and B6CBAF1 oocytes. (a) Schematic representation of the experimental design. (b) Comparison between in vitro developmental rates in MST embryos and controls. (c) Representative blastocyst images from NZB oocytes fertilized by ICSI and cultured for 96 hr (left) or MST embryos where NZB spindle was transferred into B6 strain cytoplasts (right) fertilized by ICSI and cultured for 96 hr. Note the improved blastocyst morphology upon MST. (d) In vivo development rates between MST and controls. (e) Representative neonate generated by MST with its corresponding placenta (left) and 2 day-old MST pups (right). Statistical significance was calculated with Chi-square or Fisher's exact test. *** indicates p-values<0.05.

**Table 2.** Efficiency and in vitro developmental rates of non-manipulated control and MST oocytes.

| Group | n oocytes processed by MST | | | | In vitro development for up 96 hr post-ICSI | | | |
|---|---|---|---|---|---|---|---|---|
| | n initial | Enucleated (%) | Fused (%) | ICSI survival (%) | Cultured | Two-cell (%) | Morula (%) | Blastocysts (%) |
| Control B6CBAF1 | 155 | N/A | N/A | 149 (96.1) | 144 | 144 (100.0)* | 121 (84.1)* | 112 (77.8)* |
| Control NZB | 193 | N/A | N/A | 181 (93.7) | 159 | 129 (81.1)† | 36 (22.6)† | 9 (5.6)† |
| MST *B6-St/NZB-Cyt* | 156 | 149 (95.5) | 144 (96.6) | 132 (91.7) | 110 | 93 (70.5)† | 11 (8.3)† | 0 (0.0)‡ |
| MST *NZB-St/B6-Cyt* | 270 | 238 (88.1) | 228 (95.7) | 221 (96.9) | 212 | 208 (98.1)* | 169 (79.7)* | 109 (51.4) § |

*,†,‡,§ Values with different superscripts differ significantly within the same column (p<0.05; Chi-square test or Fisher's test).

lower fertilisation and massive developmental arrest observed at preimplantation stages in the *NZB* strain (*Figure 2b,c* and *Table 2*).

At 96 hr post-insemination, embryos produced in the different experimental groups were vitrified and their competence to develop in vivo determined when synchronized pseudo-pregnant females were available for transfer. A total of 65 MST blastocysts from the *NZB-sp/B6-cyt* MST group were then warmed (100% survival) and transferred non-surgically into six recipients, which resulted in 14 live pups (21.5%) (*Figure 2d,e* and *Table 3*). This birth rate is comparable (p>0.05) with results obtained from the *B6CBAF1* control group (15 live pups (25.9%) out of 58 blastocysts transferred into five recipients). Consistent with expectations, only six pups developed to term from 44 morulas/blastocysts (13.6%) transferred into five recipients from the control *NZB* group. All living pups were born healthy and respired normally. Caesarean sections at 18.5 dpc were performed in two recipients of each group to evaluate the size and weight of the placentas and the corresponding pups, with no significant differences found between groups (*Table 4*). These results suggest that MST procedures do not typically induce an overgrowth phenotype of the type described for certain other techniques, such as somatic cell nuclear transfer (*Costa-Borges et al., 2010*). Overall, these experiments confirmed that MST, with cytoplast donation from a distantly related mouse strain, is highly effective at overcoming the in vitro developmental arrest phenotype of *NZB* mice and that the resultant embryos are competent to develop to term with high efficiency.

## mtDNA carryover analysis of biopsied cells and the complementary embryos

The extent of mtDNA carryover induced by MST was evaluated in embryos at different preimplantation developmental stages. Spindles from *NZB* oocytes were transferred into *B6CBAF1* cytoplasts and the resultant MST oocytes were fertilized by ICSI and cultured in vitro (*Figure 3a*). Afterwards, biopsies were performed to remove second polar bodies from embryos at the two-cell stage, single cells (blastomeres) from 6 to 8 cell stage embryos, or to excise a cluster of 4–8 trophectoderm cells from blastocysts (*Figure 3—figure supplement 1*). The biopsies and their corresponding embryos were then analyzed individually to ascertain whether mtDNA heteroplasmy levels in the biopsied cells are representative of the values found in the complementary embryo (*Figure 3a*).

To determine mtDNA carryover, a high-throughput sequencing protocol was developed based upon quantification of a single nucleotide polymorphism (SNP) in mtDNA using Ion PGM sequencer (ThermoFisher, see Materials and methods for further details). The SNP utilized for this purpose is

**Table 3.** In vivo developmental rates of non-manipulated control and MST oocytes.

| Group | In vivo development | | |
|---|---|---|---|
| | n transferred | n implantation sites (%) | n full-term (%) |
| Control B6CBAF1 | 58 | 23 (39.7)* | 15 (25.9) |
| Control NZB | 44 | 7 (15.9)† | 6 (13.6) |
| MST *B6-St/NZB-Cyt* | N/A | N/A | N/A |
| MST *NZB-St/B6-Cyt* | 65 | 30 (46.1)* | 14 (21.5) |

*, † Values with different superscripts differ significantly within the same column (p<0.05; Chi-square test or Fisher's test).

**Table 4.** Average weights of placentas and pups generated from control and MST oocytes.

| Group | n | Average weight | |
| --- | --- | --- | --- |
| | | Placentas (± SD) | Pups (± SD) |
| Control B6CBAF1 | 3 | 134.1 (23.3) | 802.1 (153.2) |
| Control NZB | 3 | 171.1 (27.9) | 747.9 (76.9) |
| MST *B6-St/NZB-Cyt* | N/A | N/A | N/A |
| MST *NZB-St/B6-Cyt* | 4 | 168.3 (14.1) | 923.5 (146.5) |

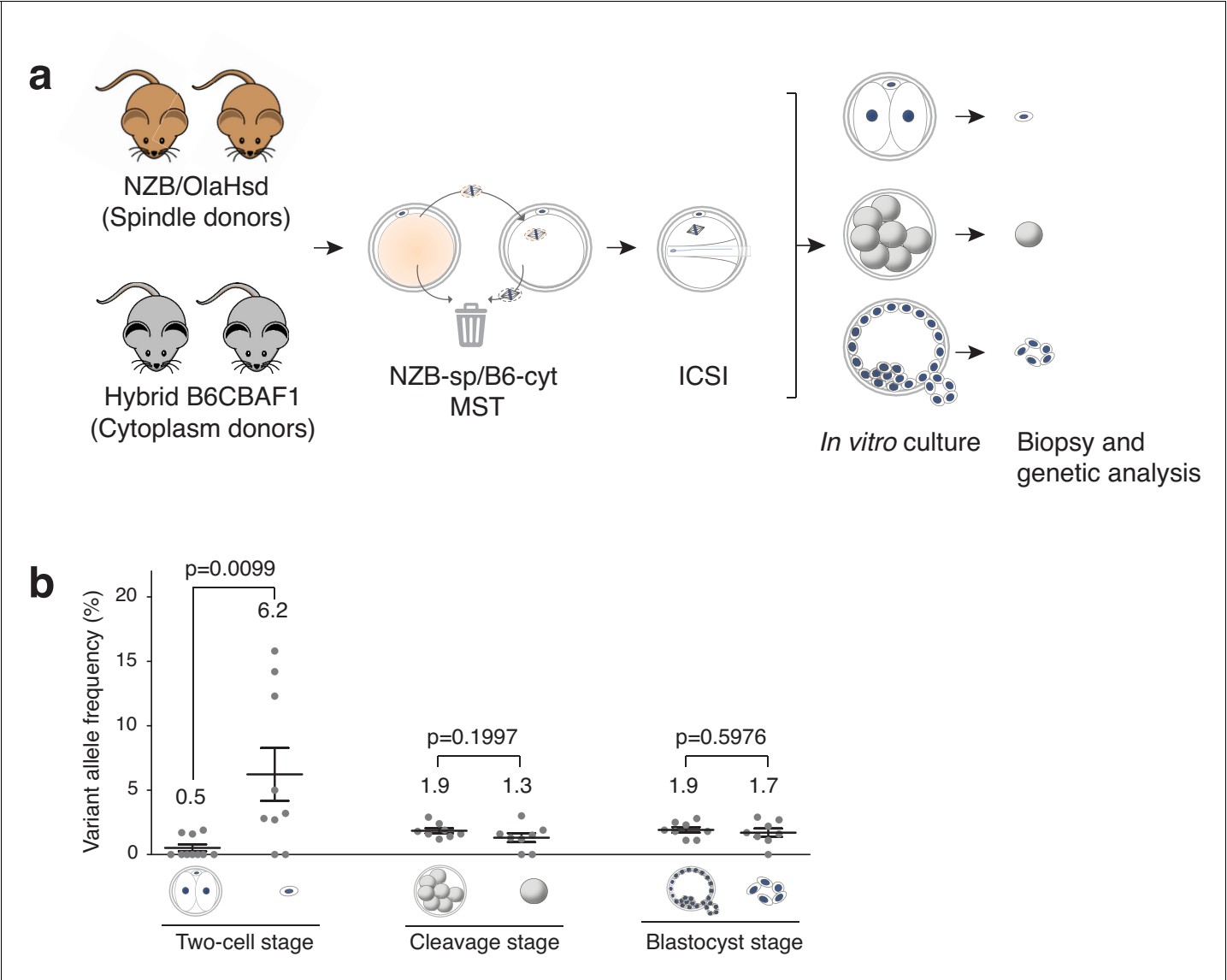

**Figure 3.** Analysis of mtDNA carryover in biopsied cells and complementary embryos from MST between NZB/OlaHsd and B6CBAF1 strain oocytes. (**a**) Schematic representation of the experimental design. (**b**) Variant allele frequencies detected in embryo specimens. Dots represent allele frequencies of individual samples. Unpaired t-test was used to compare frequencies between biopsies and corresponding entire embryos. ns = not significant. See also *Figure 3—figure supplements 1–2*.

The online version of this article includes the following figure supplement(s) for figure 3:

**Figure supplement 1.** Representative pictures of biopsy procedure in MST embryos for mtDNA heteroplasmy quantification.

**Figure supplement 2.** Validation of established high-throughput sequencing protocol for mtDNA carryover analysis.

located at position m.3932 and exists as a guanine (G) in the *B6CBAF1* strain and an adenine (A) in the *NZB* strain. The presence of different alleles at m.3932 was confirmed by minisequencing analysis using genomic DNA (gDNA) from tail tips of *B6CBAF1* and *NZB* mice (*Figure 3—figure supplement 2*). This sequencing protocol was carefully validated. Initially, protocol accuracy and sensitivity was assessed by analyzing different ratios of G and A alleles in artificially constructed samples, composed of gDNA from both mouse strains mixed in different ratios. For the purpose of these experiments, the G base (derived from B6CBAF1) was considered the reference allele and the A base (from NZB) the variant allele (*Figure 3—figure supplement 2* and *Supplementary file 1*). To verify validity of mtDNA carryover assessment by analysis of a single SNP and to ensure reliability of the utilized sequencing platform, four additional SNPs (B6CBAF1/NZB: m.2798C/T; m.2814T/C; m.3194T/C; m.3260A/G) were analyzed on a different sequencer (Illumina's MiSeq). The presence of different alleles was also confirmed by minisequencing (see Materials and methods and *Supplementary file 2* for further details; and *Figure 3—figure supplement 2*).

Analysis of mtDNA carryover after MST in biopsied cells and the complementary embryos (*Figure 3a*), revealed that the mean variant (NZB) allele frequencies obtained from polar bodies were significantly higher compared to the mean frequencies in the complementary two-cell-stage embryos (6.2 ± 6.2% SD *versus* 0.5 ± 0.8% SD; p=0.0095) (*Figure 3b*). By contrast, there was no significant difference in mtDNA allele frequencies between biopsied blastomeres and trophectoderm samples when compared to the corresponding embryos (cleavage-stage: 1.3 ± 1.0% SD *versus* 1.9 ± 0.6% SD, respectively; blastocyst stage: 1.7 ± 0.9% SD *versus* 1.9 ± 0.6% SD, respectively). Moreover, the mean heteroplasmy levels were similar between all embryonic samples (except polar bodies) (*Figure 3b* and *Supplementary file 3*). These experiments demonstrate that cleavage stage or blastocyst biopsy are preferable over biopsy of second polar bodies as methods for determining the mtDNA carryover levels found in preimplantation embryos. The results also suggest that while some mitochondria remain associated with the meiotic spindle, and are unavoidably transferred to the recipient cytoplast, the vast majority of these organelles do not persist into later developmental stages, with most being expelled into the second polar body at the completion of meiosis II.

## Developmental potential of MST mice and mtDNA heteroplasmy fate

To ascertain the long-term health status and fertility of the mice generated by MST, follow up studies were then conducted over five generations. Ten mice (three females and seven males) generated by MST were selected for mating with wild type (WT) mice. At 21 days after birth, the resultant offspring were weaned, and the size and gender ratio of the litters were assessed. All parental MST mice (F1) were fertile and produced a total of 78 pups, with a mean litter size of 7.8 ± 1.4 pups/animal and no significant deviations in the expected male-female ratio (59% and 41% respectively, *Supplementary file 4*). All pups (F2) were born alive, respired normally and grew to adulthood without manifesting any physiological or behavioral alteration. The fertility of these mice was assessed for a total of 5 generations, by selecting random males and females from litters (n = 9 in F2 and n = 4 between F3 and F5). Similarly, these mice also displayed normal fertility and produced viable offspring, without alterations in the expected gender ratio (*Supplementary file 4*).

Gross necropsies of the parents and offspring were performed during the five generations, with no pathological findings observed. In the 239 mice analyzed, all organs showed a normal size, texture and morphological appearance. Additionally, F1 mice generated by MST B6-sp/B6-cyt (n = 3), MST NZB-sp/B6-cyt (n = 5) and control B6 (n = 4) groups were also processed for histopathological examinations, which were performed in vital organs including heart, kidney, liver and brain, as well as, in tibial and quadriceps skeletal muscle and urinary bladder smooth muscle. Reproductive systems and accessory glands of both males (testis, epididymis, seminal vesicles, prostate, coagulating glands, ampullary glands and bulbourethral glands) and females (ovaries, oviducts, uterine horns) were also assessed. Except for a pericardium focal inflammation in one animal of the B6 control group, none of the animals showed any lesions or visible abnormalities (*Figure 4—figure supplements 1* and *2*). Taken together, these results support the notion that MST can efficiently produce viable and fertile offspring.

A source of great concern in MRTs field has been the reversion of mtDNA heteroplasmy observed in embryonic stem cells (ESCs) derived from pronuclear transfer or MST generated embryos (*Hyslop et al., 2016*; *Kang et al., 2016*; *Paull et al., 2013*). To evaluate whether heteroplasmy was transmitted through generations and whether homoplasmy was restored, the ratios of the mtDNA

alleles attributable to *B6CBAF1* and *NZB* were assessed through several generations. Multiple organs were assessed, including those with different metabolic demands: brain; heart; liver; kidneys (*Jenuth et al., 1997*; *Sharpley et al., 2012*). A total of six mice (four male and two female) from F1 were sacrificed at adult age (12 weeks old). The mean heteroplasmy level in this group of mice was low at 2.3 ± 1.3% (mean ± SD, n = 6) ranging from mean frequencies of undetectable values to 3.5% in individual mice (*Figure 4a*, *Supplementary file 5*). Moreover, heteroplasmy levels were similar among different tissue types from the same mouse (*Figure 4b*) and showed no differences between males and females.

Finally, the fate of the heteroplasmy was examined in adult mice derived from the MST female lineage. Four mice (two males and two females) were selected at random from each litter, through five generations. Mitochondrial DNA heteroplasmy levels were reduced to 0.4 ± 0.6% (mean ± SD, n = 4) on average in F2 mice (*Figure 4c* and *Supplementary file 5*) and decreased to undetected levels in subsequent generations (F3 to F5, *Supplementary file 5*). These quantifications based on a single SNP in an Ion PGM sequencer were corroborated by using an additional sequencing platform (Illumina's MiSeq) and 5 SNPs, as described above. Artificially constructed samples, composed of gDNA from both mouse strains mixed in different ratios, and gDNA from 5 organs of selected adult mice from F1-3 generations were analyzed (*Figure 3—figure supplement 2*, *Figure 4—figure supplement 3*, *Supplementary files 2* and *6*). These results suggest that low levels of mtDNA heteroplasmy resultant from MST typically result in a homoplasmic state in offspring within a few generations, without reversion (*Supplementary files 5* and *6*). However, it is acknowledged that different mtDNA haplogroups or mtDNA genomes affected by specific mutations might have differences in the efficiency with which they replicate, influencing the speed at which homoplasmy is attained as well as the risk if reversion.

## Discussion

MST is a technique that was originally proposed to prevent the transmission of mitochondrial diseases. This proof of concept study provides insights into the feasibility of this technique as a potential new reproductive approach to overcome infertility problems characterized by repeated in vitro embryo development arrest caused by cytoplasmic deficiencies in the oocyte.

Herein, it is shown that MST can be carried out with high efficiency in the mouse, with successful enucleation and reconstruction achieved for >95% of oocytes. Furthermore, the data produced indicate that, as long as all the steps of the protocol are well optimized and care is taken to minimize the risk of damage to the oocyte, the procedure does not negatively affect the spindle apparatus or early embryo development. In the event of a future clinical application of MST in humans, it may be difficult to coordinate the retrieval of mature oocytes from patients and donors, due to the inherent variation in ovarian responses to hormonal stimulation. For this reason, the capacity of cryopreserved oocytes to substitute for fresh oocytes, when serving as spindle or cytoplast donors, was evaluated. The results indicated that fresh and vitrified oocytes are equally suitable for use as spindle donors, but superior results are obtained if the recipient cytoplast is fresh. This agrees with a previous report performed in non-human primates that had shown that fresh spindles transplanted into vitrified cytoplasts results in impaired (50%) fertilization after ICSI, while the reciprocal spindle transfer resulted in fertilization (88%) and blastocyst formation (68%) rates similar to fresh controls (*Tachibana et al., 2009*). This also represents an advantage in the clinical setting, where low-responders to ovarian stimulation could vitrify oocytes from repeated oocyte collections, and the accumulated oocytes be used for MST using freshly collected donor cytoplasts.

Additionally, MST was conducted between two distantly related mouse strains with the aim of simulating a clinical context, in which donors with oocytes of good reproductive competence provide cytoplasts for patients with a history of poor oocyte fertilization and/or high rates of failed embryo development. The experiments demonstrated how the successful replacement of the entire cytoplasm of compromised oocytes has the potential to overcome the massive embryo development arrest phenotype, which is observed in non-manipulated controls from a sensitive mouse strain (NZB). This strategy resulted in a highly significant (10-fold) increase in blastocyst formation rates, as well as an increased likelihood of embryo development to term, compared to non-manipulated control oocytes. These results highlight the importance of the cytoplasm on the potential of the oocyte to support embryo development in vitro and to lay the foundations for a successful pregnancy.

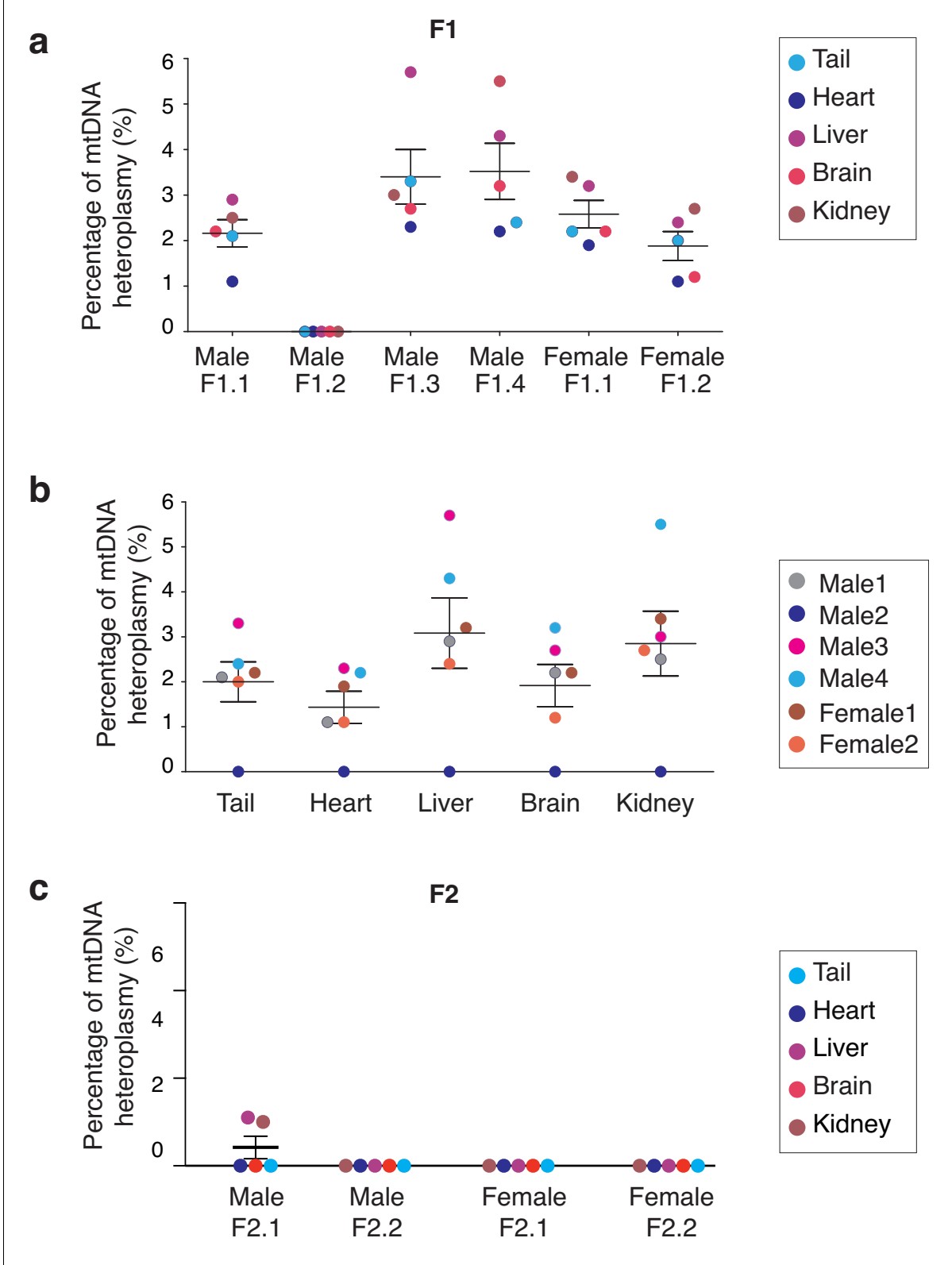

**Figure 4.** Analysis of mitochondrial heteroplasmy levels in adult mice born by MST. (a) Mitochondrial heteroplasmy levels in several organs from 4 male and two female adult MST mice (F1) are maintained below 6%. (b) Mitochodrial heteroplasmy levels are not significantly different among several organs from F1 mice (one-way ANOVA's p>0.05). (c) MST-derived mice from F2 showed undetectable levels of mtDNA heteroplasmy, except for low levels in liver and kidney in one female (F2.2). Horizontal lines represent median and standard errors of the mean. See also *Figure 4—figure supplement 3*.
*Figure 4 continued on next page*

*Figure 4 continued*

The online version of this article includes the following figure supplement(s) for figure 4:

**Figure supplement 1.** Hematoxylin and eosin (H and E) stained sections of adult mice generated from B6 ICSI control embryos, B6 reciprocal MST and NZB-sp/B6-cyt MST embryos.

**Figure supplement 2.** Hematoxylin and eosin (H and E) stained sections of the reproductive organs of adult mice generated from B6 ICSI control embryos, B6 reciprocal MST and NZB-sp/B6-cyt MST embryos.

**Figure supplement 3.** Validation of established sequencing protocol for mtDNA carryover analysis using two sequencing platforms.

---

Consistent with this data, Mitsui and colleagues showed that oocyte genomes from mice aged 10–12 months transferred into oocytes of young mice aged 3–5 months, resulted in increased term-development from 6.3% for in vivo aged oocytes to 27.1% for the reconstructed oocytes (*Mitsui et al., 2009*). Similarly, a recent study demonstrated that in vitro aged oocytes accumulate cytoplasmic deficiencies if they are maintained in culture for an extended period prior to fertilization, and that these deficiencies can be overcome with spindle transfer (*Yamada and Egli, 2017*). However, in both cases, studies were performed between oocytes from the same mouse strain and thus the potential of the technique to overcome infertility in a strain with poor fertility competence remained undetermined. It is noteworthy that the current study employed ICSI to inseminate oocytes, rather than conventional IVF, which resembles closely the standard protocol used in humans for oocytes that have been denuded of the surrounding cumulus cells.

Levels of mtDNA heteroplasmy caused by carryover of mitochondria in close proximity to spindle during the MST procedure were also evaluated. Clearly, this is an important consideration when utilizing MST technology to avoid transmission of mtDNA mutations responsible for serious inherited disorders, but it is also relevant to other variations in the mtDNA, or other defects affecting the mitochondrial organelle, which may potentially contribute to certain forms of embryonic developmental arrest. As well as assessing the extent of heteroplasmy at different embryonic stages, the levels were also assessed in multiple tissues in adulthood and over several generations. There are conflicting reports regarding the dynamics of mtDNA heteroplasmy during the lifetime of an individual, between organs or even during in vitro culture when ESCs have been derived from MRT embryos with heteroplasmic mtDNA (*Hyslop et al., 2016*; *Kang et al., 2016*; *Paull et al., 2013*). It is also unclear to what extent divergent mtDNA haplotypes in heteroplasmic organisms might lead to functional incompatibility, either between the two types of mitochondria or between the mitochondrial and nuclear genomes.

This study confirms that cells biopsied from MST embryos at the morula or blastocyst stages present minimal levels of heteroplasmy (<2.9% mtDNA from the spindle donor) and that these biopsy specimens are representative of the remainder of the embryo. On the contrary, the heteroplasmy levels were significantly higher in second polar bodies than in blastomere or trophectoderm biopsies. This agrees with data from Neupane and colleagues, who have shown that, in comparison to second polar bodies, mtDNA heteroplasmy in TE cells is more closely correlated with the levels in the blastocyst as a whole or the corresponding ESCs (*Neupane et al., 2014*). Alternatively, oocytes might be actively removing mitochondria transferred along with the spindle, since they may be disadvantageous as compared to the recipient's own organelles (*De Fanti et al., 2017*). However, perhaps the most likely explanation is that when the meiotic spindle is transferred, a number of mitochondria accompany it. These mitochondria are likely to remain in the vicinity of the MII spindle and consequently it is inevitable that a disproportionate number of these mitochondria will pass into the second polar body. Regardless of the underlying mechanism, our data suggest that testing of blastomeres or TE biopsies is preferable to second polar body analysis for the quantification of mtDNA heteroplasmy levels. This also has relevance for the preimplantation genetic testing (PGT, also known as preimplantation genetic diagnosis – PGD) of mitochondrial disease in the human.

The data from our current study revealed that mtDNA heteroplasmy levels were low in all the adult mice produced, regardless of gender, or the type of organ (range 0–6%). Previous studies in monkeys and humans have shown that a minimal number of donor mitochondria are transferred using MRT (below 1–2%) (*Craven et al., 2010*; *Hyslop et al., 2016*; *Paull et al., 2013*; *Tachibana et al., 2009*). Nevertheless, since the meiotic spindle in mouse oocytes is much larger than that of the human, and given that multiple mitochondria are found in the vicinity of the spindle,

it was expected that MST in mice would lead to higher mtDNA carryover levels. The surprisingly low heteroplasmic levels achieved during this study can likely be attributed to the use of birefringence microscopy during enucleation, which assists in minimizing the carryover of cytoplasm transferred along the meiotic spindle.

It has also been suggested that organs with a high-metabolic demand tend to accumulate higher heteroplasmy mtDNA levels (*Jenuth et al., 1997*; *Meirelles and Smith, 1997*), however, our data do not confirm this observation. This result could be explained by differences in mitochondrial haplotypes from mouse strains used, which could have a differential replication rate.

Some studies of heteroplasmic ESC lines derived from embryos carrying mtDNA mutations have shown changes in the levels of normal and mutant mtDNA during prolonged in vitro culture, with reversion back to a situation where mutant mtDNA predominates (*Hyslop et al., 2016*; *Kang et al., 2016*; *Paull et al., 2013*). This delayed efforts for the direct application of MRT-derived techniques in the clinical setting and raises some concerns for the first baby born using MST (*Zhang et al., 2017*). Nevertheless, the results presented here show a low level of mtDNA carryover in all adult organs analyzed, suggesting that the mechanism seen in ESCs in vitro might not necessarily represent the in vivo process. The results also agree with Sharpley et al, who showed that NZB and 129S6 mtDNA heteroplasmic haplotypes decrease over generations (*Sharpley et al., 2012*). In the current study, heteroplasmy was very low in the F2 progeny and undetected in the offspring of the subsequent generations (up to F5). The data collected from the analyzed organs suggests that heteroplasmy resultant from MST can be stable within an individual and can lead to an homoplasmic state within a few generations. However, additional work should be done in order to comprehensively assess how mtDNA heteroplasmy segregates in other organs.

On the other hand, the MST mice followed over five generations were apparently normal and showed good fertility (average of 7.8 pups per litter). This is a notable observation, as based on the literature, NZB/OlaHsd mice are expected to have small litter sizes (3.8 at weaning) (*Fernandes et al., 1973*; *Hansen CT and Whitney, 1973*). Additionally, histological examinations in F1 MST mice did not reveal any lesions in a selection of organs. Whether the MST technique can potentially reveal mitochondrial causes of infertility that are hereditary or aggravated with lifestyle or age is a question that remains to be answered and will require additional studies.

In conclusion, this study has demonstrated that MST can overcome a severe developmental arrest phenotype, associated with poor fertility and greatly reduced chances of an individual oocyte producing a pregnancy following in vitro fertilization. The results show that embryos produced using optimized MST techniques can give rise to apparently normal and fertile animals. Levels of heteroplasmy were low in the initial generation and undetectable in subsequent generations, indicating that homoplasmy for the mtDNA of the cytoplast donor is rapidly attained in this model. Given the high proportion of IVF cycles which are unsuccessful due to poor embryo development related to low oocyte quality, we believe that there is a need to further explore the potential of MST as a clinical treatment for infertility. Pre-clinical and clinical trials involving human oocytes, undertaken in a regulated and carefully controlled manner, is desirable, since such a therapy could represent the last chance for infertile patients to have genetically related children.

## Materials and methods

### Mice

Animal care and procedures were conducted according to protocols approved by the Ethics Committee on Animal Research (DAMM-7436) of the *Parc Cientific of Barcelona* (PCB), Spain. Hybrid (*B6/CBA*) and outbred *CD1* females of 5–6 weeks of age (25–30 g), and male mice from the same genetic strains of 8–10 weeks of age (25–30 g) were purchased from Janvier Laboratories (France). *New Zealand Black* (*NZB/OlaHsd*) mice were purchased from Envigo (France). Upon arrival, all mice were quarantined and acclimated to the PCB Animals´ facility (PRAL) for approximately 1 week prior to use. Three to four mice were housed per cage in a room with a 12 hr light/dark cycle with ad libitum access to food and water.

## Oocytes and sperm collection

For the collection of oocytes, hybrid *B6CBAF1* and *NZB* females were induced to superovulate by intraperitoneal injection of 5 IU of pregnant mare serum gonadotropin (PMSG) followed 48 hr later by 5 IU of human chorionic gonadotropin (hCG). Cumulus–oocyte complexes from the both strains were released from the oviducts by 14–15 hr after hCG administration and treated with hyaluronidase (LifeGlobal) until cumulus cells dispersed. Once denuded, oocytes with good morphology were washed several times and kept in culture medium (Global total, LifeGlobal) under oil (Lifeguard, LifeGlobal) at 37.3°C, in an atmosphere with 7%CO2% and 7%O2 in air, until use. Sperms were collected from cauda epididymis and then diluted and incubated in medium supplemented with glucose (Global total for fertilization, LifeGlobal) at 37.3°C, in an atmosphere with 7%CO2% and 7%O2 in air, until use.

## Spindle transfer, ICSI and embryo culture

Oocytes from *B6CBAF1* or *NZB* strains were used as spindle chromosome-complex and cytoplasts donors. Procedures were performed using a piezo-driven (PiezoXpert, Eppendorf) micromanipulator. Oocytes first were exposed to small drops of hepes-buffered medium (Global total w/hepes, LifeGlobal) containing 5 µg/mL cytochalasin B (Sigma) covered with mineral oil for 3–5 min at 37°C. Afterwards, the meiotic spindle was aspirated into an enucleation pipette (Humagen) trying to remove the minimum amount of surrounding cytoplasm possible, and enucleation confirmed using a microtubule birefringence system (PolarAide, Vitrolife) to visualize the spindle apparatus (*Figure 1— figure supplement 1*). If the karyoplast removed contained a larger amount of cytoplasm, the extra cytoplasm was eliminated by pressing the cytoplasm against the zona pellucida. Karyoplasts were inserted below the zona pellucida of another enucleated oocyte (cytoplast) and fused using inactivated Sendai virus HVJ-E (GenomeOne, Cosmo Bio). All manipulations were performed on a 37°C heated stage (Okolab) of an Olympus IX73 inverted microscope, using Eppendorf micromanipulators. Non-manipulated control oocytes and those generated by MST were inseminated using a modified piezo-actuated ICSI technique, known as the 'hole removal technique'. Briefly, this procedure is based on withdrawing the ICSI pipette and applying rapid suction simultaneously just after the sperm head has been injected to seal the oocyte membrane, which increases survival chances. The injected oocytes were then cultured in Global total medium (LifeGlobal) under oil at 37.3°C, in *K-Minc* incubators (Cook Medical), in an atmosphere with 7% CO2 and O2 in air.

## Embryo biopsy and tubing of cells for molecular analysis

Embryos generated by MST were biopsied at different developmental stages, including: two-cell, morula or blastocyst stage. Regardless of the developmental stage, biopsies were performed in individual 5 µL droplets of Global total w/hepes medium covered with oil using a biopsy pipette with 19 µm of internal diameter (Eppendorf) with the assistance of laser shots to open a hole in the zona pellucida or to weaken the trophectoderm cells in the case of the blastocyst biopsy. After biopsy, both the biopsied cells and the complementary embryo were transferred individually to empty PCR tubes and stored at −80°C until processed for mtDNA allele frequencies determination.

## Fluorescence analysis

For analysis of the spindle structure and chromosomes distribution, control and MST oocytes were fixed and extracted for 30 min at 37°C in a microtubule stabilizing buffer (MTSB-XF). A triple-labeling protocol was then used for the detection of microtubules, microfilaments and chromatin by fluorescence microscopy, as described previously (*Messinger and Albertini, 1991*). Briefly, fixed oocytes were first incubated in a mixture of mouse monoclonal anti α/β-tubulin antibodies, and then in a mixture of secondary antibody (chicken anti-mouse IgG) conjugated to Alexa Fluor 488 and of Alexa Fluor 594 phalloidin. Finally, all oocytes were washed in PBS blocking solution, incubated in Hoechst 33258, and put on a mounting solution droplet on a glass slide. Blastocysts processed for total cell counts were fixed in 4% PFA and permeabilized in 2.5% Triton-X100 for 25 min at room temperature. Afterwards, blastocysts were incubated overnight in blocking solution and then in rabbit monoclonal anti-Oct-4, washed 3 times in PBS blocking solution for 10 min at 37°C. After, they were incubated in secondary antibody (goat anti rabbit IgG) conjugated with Alexa Fluor 594, washed and incubated in Hoechst (10 µg/ml) for 10 min at room temperature and finally mounting solution

droplet on a glass slide. Stained oocytes or blastocysts were examined using an epifluorescence microscope (Nikon E1000) fitted with specific filters for Hoechst, Fluorescein and Texas Red and a 50W mercury lamp. Digital images were acquired with E1000 Nikon software.

## Oocyte and blastocyst vitrification and transfer

Oocytes and blastocysts were vitrified following the instructions provided by the manufacturer (Kitazato BioPharma Japan). Briefly, samples were exposed to equilibration solution (ES) for 15 min, transferred to VS1 for 30 s and then to VS2 for additional 30 s. Afterwards, they were loaded onto the surface strip of a classic Cryotop (Kitazato BioPharma Japan) and directly plunged into liquid N2. For warming, the Cryotop strip was transferred from the liquid nitrogen into a TS solution for 1 min at 37°C and then gradually moved to dilution solution (DS) for 3 min, to washing solution (WS) 1 for 5 min and, finally, to WS2 for an additional 1 min. Exposures to DS and WS solutions were performed at room temperature. After warming, samples were extensively washed and kept in culture medium under oil at 37.3°C, in an atmosphere with 7% $CO_2$ and $O_2$ in air.

## Embryo transfer

Embryo transfers were performed non-surgically using a commercial non-surgical embryo transfer protocol (NSET, Paratechs). Briefly, an NSET device was coupled to a P2 pipette with volume adjusted to 1.8 µl. Between 8 and 12 blastocysts were loaded in each device within a culture medium droplet under a stereomicroscope. After loading the blastocysts, the volume in the P2 pipette was re-adjusted to 2 µl to create an air bubble and to avoid the loss of the embryos by capillarity. The recipient female assigned for transfer was then immobilized, and a NSET small speculum was carefully introduced in the vagina. With the animal still immobilized, the NSET device loaded with the embryos was introduced by the speculum through the cervix. When the base of the device got in contact with the speculum, the blastocysts were transferred by pressing the plunger of the pipette. Having the plunger of the pipette still pressed, NSET device was removed and checked under the stereomicroscope to confirm that all embryos had been correctly transferred. Finally, the speculum was removed and the female returned to its corresponding cage.

## Birth control and follow-up of the offspring

In the majority of transferred females natural delivery was controlled at the day 20 of pregnancy (P20), while in a few cases, cesarean sections were performed on embryonic day 18.5 to collect information on the weight and size of the placentas and pups. Pups (F1) resultant from the embryo transfer procedures were checked for health status and grown up until sexual maturity age was reached. Having reached the adult age, F1 males and females from each experimental group were randomly selected for crossing with wild-type (WT) *B6CBAF1* mice, so that their health status and fertility competency could be assessed. At day 21 after birth, the offspring of the F1xWT = F2 mice were weaned and the F2 animals were checked and sexed. The same strategy was repeated for a total of 5 generations, by selecting random males and females from litters (n = 9 in F2 and n = 4 between F3 and F5).

## Histological analysis

For histological evaluation, tissue samples from 4 ICSI-B6 control, 3 B6-sp/B6-cyt MST and 5 NZB-sp/B6-cyt MST mice were collected at 6 weeks of age. Mice were perfused with PBS and 5% formaldehyde solution. Subsequently, tissues were fixed overnight at 4°C in 5% formaldehyde and embedded in paraffin wax, sliced in 4 µm sections and stained with hematoxylin and eosin staining (H and E). The atrium, valves and myocardium of heart, kidney, liver and gall bladder, forebrain, midbrain and hindbrain, tibial and quadriceps muscle, urinary bladder and reproductive organs (testis, epididymis, accessory glands, ovary and uterus) were evaluated. Histological analysis was carried out blindly using mouse identification codes for group assignment that were unknown to the evaluator.

Analysis of mitochondrial DNA carryover mtDNA carryover in embryo specimens and adult mouse tissues was determined by SNP quantification using a high-throughput sequencing protocol. Prior to sequencing, polymerase chain reaction (PCR) was performed to amplify the SNP located at m.3932 in the mtDNA (B6CBAF1: A; NZB: G). DNA from embryo specimens was obtained by alkaline lysis. After the addition of 0.75 µl nuclease-free water, 1.25 µl 0.1M DL-Dithiothreitol and 0.5 µl 1.0M

Sodium hydroxide solution (per sample), cells were lysed at 65℃ for 10 min. Genomic DNA (gDNA) from organs (tail tips, hearts, brains, livers and kidneys) was extracted using the DNeasy Blood and Tissue Kit from Qiagen. A single PCR mixture consisted of 1.5 µl HotMaster Taq DNA Buffer with Magnesium (5 Prime), 0.6 µl of 100 µm primer pool (5'-CCATACCCCGAAAACGTTGG-3' and 5'-GG TTGGTGCTGGATATTGTGA-3'), 0.3 µl 10 nM dNTP Mix and 0.09 µl HotMaster Taq DNA Polymerase (5 Prime). The PCR mix was added to lysed embryo specimens along with 7.99 µl nuclease-free water and 2.5 µl 0.4M Tricine (per sample) and to 0.5 µl of gDNA along with 12.49 µl nuclease-free water (per sample). PCRs were performed using the following conditions: 96.0℃ for one minute; 35 cycles of 94.0℃ for 15 s, 58℃ for 15 s and 65.0℃ for 45 s; 65.0℃ for two minutes. Successful amplification was verified by gel electrophoresis. Sequencing libraries were prepared from PCR amplicons using the Ion Plus Fragment Library Kit from ThermoFisher. Libraries were sequenced on the Ion Personal Genome Machine (PGM; ThermoFisher). The Torrent Variant Caller plugin (ThermoFisher) was used for SNP allele quantification. In order to increase variant calling accuracy the settings for 'Somatic' variants were set to 'High Stringency', to enable low frequency variant detection at a minimal false-positive call rate. The read depth was downsampled to 20,000 to increase accuracy of variant calls. A 'HotSpot Region' BED file, defining the exact genomic coordinate of the assessed nucleotide, in addition to a 'Target Region' BED file, was used.

Prior to the analysis of embryo specimens and tissue samples, validation experiments were performed. Minisequencing was used to confirm SNP alleles A and G at position m.3932 in B6CBAF1 and NZB mouse strains, respectively. PCR amplicons (1 µl) from gDNA (extracted from tail tips) of the B6CBAF1 and NZB mouse strains were treated with 0.5 µl EXOSAP-it (Affymetrix) and incubated at 37℃ for 15 min and 80℃ for 15 min. PCR amplicons (1.5 µl) were combined with 0.5 µl water, 2.5 µl SNaPshot Multiplex Ready Reaction Mix (ThermoFisher) and 0.5 µl primer (2 µM; 5'-AATAAATCC TATCACCCTT-3'; 5'-ATTGTGAAGTAGATGATGG-3'). Mixtures were incubated using the following conditions: 25 cycles of 96.0℃ for ten seconds, 50℃ for 5 s and 60℃ for 30 s. Products were analyzed by capillary electrophoresis on a genetic analyser (ThermoFisher). The resulting data were analyzed using GeneMapper v4.0 software (Applied Biosystems). DNA mixing experiments were performed to ensure accuracy and sensitivity of the SNP quantification protocol. Sample mixtures were created by combining gDNA (extracted from tail tips) from both mouse strains at different ratios (B6CBAF1/NZB: 100/0; 98/2; 96/4; 94/6; 92/8; 90/10; 75/25; 50/50; 25/75; 0/100). Samples were sequenced and obtained ratios compared to those expected.

To ensure both the validity of assessment of a single SNP for mtDNA carryover analysis and the accuracy of the utilized sequencing platform (Ion PGM); a second set of experiments was performed, which included analysis of four additional SNPs (B6CBAF1 > NZB: m.2798C > T; m.2814T > C; m.3194T > C; m.3260A > G) utilizing Illumina's MiSeq System sequencing platform. Minisequencing was used to confirm presence/absence of SNPs in B6CBAF1 and NZB mouse strains. PCR and minisequencing procedures were performed as described above. In brief, PCR was performed to amplify additional SNPs in gDNA from B6CBAF1 and NZB tail tips (m.2798 and m.2814: 5'-AACACTCCTCG TCCCCATTC-3; and 5'-TGGACCAACAATGTTAGGGC-3'; m.3194 and m.3260: 5'-GCCGTAGCC-CAAACAATTTC-3' and 5'-GGTCAGGCTGGCAGAAGTAA-3'). Amplicons were subjected to minisequencing with following primers: 5'-TCGTCCCCATTCTAATCGC-3' and 5'-TGTTAGGAAGGCTAT-3' (m.2798T); 5'-ATAGCCTTCCTAACA-3' and 5'-AAGATTTTGCGTTCTACTA-3' (m.2814); 5'-ATGAAG TAACCATAGCTAT-3' and 5'-ATAGAACTGATAAAAGGAT-3' (m.3194); 5'-CACTTATTACAACC-CAAGA-3' and 5'-GCAGAAGTAATCATATGTG-3' (m.3260). Sequencing libraries were prepared from PCR amplicons using the TruSeq DNA Nano LT kit from Illumina and sequenced on the MiSeq System. Analysis was performed with Miseq Reporter and Illumina's Somatic Variant Caller. Again, gDNA mixtures (B6CBAF1/NZB: 100/0; 99/1; 97/3; 90/10; 75/25; 50/50; 0/100) were sequenced and obtained ratios compared to those expected. Furthermore, gDNA samples from organs of three individual mice were sequenced and results compared to those obtained by PGM sequencing.

## Quantification and statistical analysis

Experiments involving micromanipulation procedures were usually repeated between 6 to 9 times. Results obtained in the different replicates were pooled and analyzed together. Oocytes used for manipulation were always taken randomly from a common pool of oocytes collected from the 4–6 female mice used on each experimental day. In all experiments that involved embryo culture, control groups with non-manipulated oocytes were always processed and cultured in parallel together with

the manipulated groups. All statistical analyses were performed using Prism 6.0 program (Graph-Pad). For comparisons of mean cell numbers, placentas and mice weights, mtDNA carryover values in embryo species and adult mouse tissues, a *t-test* or one-way *ANOVA* was performed, where the significance was set at $p < 0.05$. For the analysis of oocyte/embryo proportions, chi-square test was performed and a p-value $< 0.05$ was considered significant.

- Validation of established sequencing protocol for mtDNA carryover analysis measured by Ion PGM sequencer. Allele frequencies at position m.3932 in homoplasmic samples and artificially constructed heteroplasmic sample mixtures. See also *Source data 1*.

## Acknowledgements

This study was financed by European Regional Development funds (ERDF) Ref RD 15-1-0011 conceded to Embryotools. DW is supported by the NIHR Oxford Biomedical Research Centre and National Institutes of Health grant 1R01HD092550-01.

## Additional information

### Funding

| Funder | Grant reference number | Author |
| --- | --- | --- |
| European Regional Development Fund | RD-15-1-0011 | Nuno Costa-Borges |
| National Institute for Health Research | 1R01HD092550-01 | Dagan Wells |

The funders had no role in study design, data collection and interpretation, or the decision to submit the work for publication.

### Author contributions

Nuno Costa-Borges, Conceptualization, Data curation, Supervision, Validation, Investigation, Methodology, Project administration; Katharina Spath, Conceptualization, Data curation, Investigation, Methodology; Irene Miguel-Escalada, Conceptualization, Data curation, Formal analysis, Supervision, Validation, Investigation, Methodology; Enric Mestres, Maria Garcia-Jiménez, Data curation, Investigation, Methodology; Rosa Balmaseda, Ivette Vanrell, Jesús González, Investigation, Methodology; Anna Serafín, Investigation; Klaus Rink, Resources, Validation, Methodology; Dagan Wells, Conceptualization, Supervision, Funding acquisition, Methodology, Project administration; Gloria Calderón, Conceptualization, Resources, Supervision, Funding acquisition, Project administration

### Author ORCIDs

Nuno Costa-Borges https://orcid.org/0000-0002-2073-7515
Irene Miguel-Escalada http://orcid.org/0000-0003-3461-6404
Enric Mestres http://orcid.org/0000-0001-6140-6416
Maria Garcia-Jiménez http://orcid.org/0000-0003-3321-8869
Gloria Calderón http://orcid.org/0000-0003-3235-0323

### Ethics

Animal experimentation: Animal care and procedures were conducted according to protocols approved by the Ethics Committee on Animal Research (DAMM-7436) of the Parc Cientific of Barcelona (PCB), Spain.

### Decision letter and Author response

Decision letter https://doi.org/10.7554/eLife.48591.sa1
Author response https://doi.org/10.7554/eLife.48591.sa2

## Additional files

### Supplementary files
- Source data 1. Variant site coverages referring to *Supplementary file 1*.
- Source data 2. Variant site coverages referring to *Supplementary file 2*.
- Source data 3. Variant site coverages referring to *Supplementary file 3*.
- Source data 4. Variant site coverages referring to *Supplementary file 4*.
- Source data 5. Variant site coverages referring to *Supplementary file 6*.

- Supplementary file 1. Validation of established sequencing protocol for mtDNA carryover analysis measured by Ion PGM sequencer. Allele frequencies at position m.3932 in homoplasmic samples and artificially constructed heteroplasmic sample mixtures. See also *Source data 1*.

- Supplementary file 2. Validation of established sequencing protocol for mtDNA carryover analysis using MiSeq sequencer. Allele frequencies at positions m.2798, m.2814, m.3194, m.3260 and m.3932 in homoplasmic samples and artificially constructed heteroplasmic sample mixtures. See also *Source data 2*.

- Supplementary file 3. mtDNA heteroplasmy analysis of biopsies and complementary embryos using Ion PGM platform. Allele frequencies at position m.3932 are shown. See also *Source data 3*.

- Supplementary file 4. Analysis of fertility and developmental potential of MST litters through five generations.

- Supplementary file 5. Allele frequencies at position m.3932 in adult MST mouse tissues through five generations using Ion PGM platform. See also *Source data 4*.

- Supplementary file 6. Validation of established sequencing protocol for mtDNA carryover analysis using MiSeq platform. Allele frequencies at positions m.2798, m.2814, m.3194, m.3260 and m.3932 in adult MST mouse tissues. See also *Source data 5*.

- Transparent reporting form

### Data availability
All data generated or analysed during this study are included in the manuscript and supporting files.

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
