## [Decision Letter]

Thank you for submitting your article "Maternal spindle transfer overcomes embryo developmental arrest caused by ooplasmic defects in mice" for consideration by *eLife*. Your article has been reviewed by three peer reviewers, and the evaluation has been overseen by a Reviewing Editor and Didier Stainier as the Senior Editor. The reviewers have opted to remain anonymous.

The reviewers have discussed the reviews with one another and the Reviewing Editor has drafted this decision to help you prepare a revised submission.

Summary:

In this manuscript, the authors present results of their study on the efficacy of cytoplasmic replacement in oocytes using mitochondrial spindle transfer (MST) for overcoming poor preimplantation development in the mouse model. MST can be performed after oocyte vitrification and frozen/thawed cytoplasts but not spindles show poor development after MST. Whole cytoplasmic replacement in mature oocytes from NZB/OlaHsd mice with donor cytoplasm derived from B6CBAF1 mice can rescue poor fertilization, cleavage and blastocyst formation typical for NZB mice.

Essential revisions:

1) Introduction and summary of the literature: "The introduction is very selective especially related to autologous mitochondrial supplementation and appears to have been crafted to sell the authors' point of view. There are data to suggest that mitochondrial supplementation works and the effect that is has on gene expression in embryos and on litter size. Furthermore, children have been produced using this approach. The authors are requested to ensure that there is scientific balance in the Introduction for each aspect discussed." Please revise.

2) "There is no detail about the depth of the sequencing. Ion Torrent is not a reliable sequencing platform – it produces too many unreliable reads – you need to use a more reliable platform such as MiSeq. The levels of heteroplasmy need to be assessed on a range of tissues to determine if you see uniform similarities in transmission. At the moment, these data are not convincing as too few tissues are assessed."

"Conclusions on mtDNA heteroplasmy are weak because results are based on detection of only one SNP difference between NZB/OlaHsd and B6CBAF1 strains. It is likely that mtDNA haplotypes of these two strains differ on more than 90 SNPs. Therefore, authors should monitor heteroplasmy on each of these positions. This should be done by whole mtDNA sequencing, which is required for conclusive heteroplasmy assays post MRT."

Please provide additional sequencing data to solidify this key point.

3) "All pups (F2) were born alive, respired normally and grew to adulthood without manifesting any physiological or behavioral alteration.” and “Gross necropsies of the parents and offspring were also performed during the 5 generations, with no pathological findings observed.” Where are these data and how much depth was there to these analyses? This is critical given that a recent paper has shown that autologous mitochondrial supplementation can lead to a heart defect. Clarify to what depth mice were studied at autopsy.

4) "Authors used NZB strain as an example of a poor breeder strain. Strikingly, F1 females derived from MST that are still genetically NZO produced large litters (~8 pups) according to Supplementary file 1. It seems that authors show only one litter, which seems to be large even in unmanipulated NZO strain. It would be useful to show if F1 females maintain high productivity over time compared to NZO females (which should be also included in the table). If MST transfer improved NZO female's fecundity this could indicate that their breeding problems are mitochondria related and replacing mitochondria could improve the performance of this strain. Which would be of interest also from the clinical perspective. This technique could potentially reveal mitochondrial causes of infertility in patients which can be hereditary in the female lineage Women treated for ooplasm deficiency could pass this to their daughters." Please comment on the reviewer's observation.

[Editors' note: further revisions were suggested prior to acceptance, as described below.]

Thank you for resubmitting your work entitled "Maternal spindle transfer overcomes embryo developmental arrest caused by ooplasmic defects in mice" for further consideration at *eLife*. Your revised article has been favorably evaluated by Didier Stainier (Senior Editor), a Reviewing Editor, and three reviewers.

The manuscript has been improved but there are some remaining issues that need to be addressed before acceptance, as outlined below:

Reviewers remain concerned about the strength of the evidence to show that the animals were indeed normal. Were any organs preserved for histological examination? If so, it would be reasonable to examine at least a subset of these. The reviewers feel that given the impetus to take this technology to the clinic, this information from your rather extensive preclinical study is of considerable importance. Any additional data related to this point would be most valuable.

Reviewer #1:

Whilst I am not convinced by the explanation for the use of Ion Torrent, it is not my primary concern.

I am primarily concerned about the lack of detail related to the health and well-being of the offspring. The subsection “Developmental potential of MST mice and mtDNA heteroplasmy fate” does not provide sufficient information to determine if the animals were healthy or not and no data are presented. There is also no detail in the Materials and methods section. Macroscopic and microscopic investigations need to be performed. Currently, there is a drive to push this form of assisted reproductive technology into clinical practice in various countries and the real question relates to the health and well-being of the offspring. A number of studies have offered accounts of the fate of mtDNA that is carried over. Your study requires some novelty and macroscopic and microscopic investigations would provide this.

I am not prepared to support the publication of a manuscript related to this form of assisted reproductive technology that does not deal with the crucial question about the health and well-being of the offspring. Three lines of text is simply not sufficient.

Reviewer #2:

Authors provided additional information requested by reviewers such as references, added text and supplementary data. Other comments were addressed in the rebuttal by providing references supporting their argument but without adding additional data (although cited papers did not always agree – see point 1).

I am satisfied with most of the issues addressed but would like to hear from other reviewers if their comments were sufficiently addressed.

1) Addressing comment 2 that 5 tissues are insufficient to determine transmission of heteroplasmy authors cited two papers (Sharpley et al., 2012; Jenuth et al., 1997) saying that they have tested "a similar number" of tissues when in fact they used twice as many (10 or 9).

Reviewer #3:

I believe that the revision did not address the main concern expressed by the reviewers that whole mtDNA sequencing (MiSeq) is required to validate conclusions on heteroplasmy. It is standard now to validate sequencing by two independent approaches. The author's arguments in the rebuttal are not convincing.

[Editors' note: further revisions were suggested prior to acceptance, as described below.]

Thank you for resubmitting your work entitled "Maternal spindle transfer overcomes embryo developmental arrest caused by ooplasmic defects in mice" for further consideration by *eLife*. Your revised article has been evaluated by Didier Stainier (Senior Editor) and a Reviewing Editor.

The manuscript has been improved but there are some remaining issues that need to be addressed before acceptance.

You will see that one reviewer continues to have concern over the scope of your assessment. Although these concerns stem from genuine concern over the interpretation of the study, we think that you have gone some way to address this with additional data. Therefore, some comment in the Discussion to indicate that additional work would be required to fully assess safety issues.

Reviewer #1:

I am pleased to see that the authors have taken on extra work to seek to validate their claims regarding the safety of MST.

I note that the authors have undertaken the histopathology analysis. However, I am surprised to see that they have restricted themselves to four organs. Likewise, as argued by one of the other reviewers, they do not carry out mtDNA analysis on all tissues. Therefore, the arguments they present are only partially valid. The reason for this is that mtDNA, and specifically mtDNA mutations (as in disease) or variants (in non-pathological situations), do not segregate neutrally, or evenly, amongst all the tissues / organs but rather in a random manner. Therefore, without having analysis of all tissues and organs (or at least 10 as agreed by another reviewer), the outcomes are only indicative of what has been analysed. I, further, note that muscle has not been analysed and myopathies are associated with mtDNA disease and, in the disease state, muscle often carries high loads of mutant mtDNA, which affects OXPHOS function (mtDNA encodes for key genes of the electron transfer chain that performs OXPHOS).

In conclusion, I think this is an important study but it is being held back though either reluctance or a lack of understanding of mtDNA genetics and, specifically, transmission and segregation of the mitochondrial genome. Either way, this results in the conclusions being inconclusive.

---

## [Author Response]

Essential revisions:1) Introduction and summary of the literature: "The introduction is very selective especially related to autologous mitochondrial supplementation and appears to have been crafted to sell the authors' point of view. There are data to suggest that mitochondrial supplementation works and the effect that is has on gene expression in embryos and on litter size. Furthermore, children have been produced using this approach. The authors are requested to ensure that there is scientific balance in the Introduction for each aspect discussed." Please revise.

We agree that the original Introduction as submitted was overly brief in describing previous studies on mitochondrial supplementation. This was due to the need to fully convey essential details to introduce our hypothesis, while keeping the text as succinct as possible. As stated by the reviewers, there are multiple studies in animal models that have described benefits of the technique to treat infertility (Yi et al., 2007; El Shourbagy et al., 2006; Hua et al., 2014) and, importantly, children resultant from the application of this technique have been born (Fakih et al., 2015). We have rewritten this section of our Introduction to better introduce autologous mitochondrial supplementation.

The following references have also been added to the current version of the manuscript:

Johnson, J., Canning, J., Kaneko, T., Pru, J.K., and Tilly, J.L. Germline stem cells and follicular renewal in the postnatal mammalian ovary. Nature. 2004; 428: 145–150.

White, Y.A., Woods, D.C., Takai, Y., Ishihara, O., Seki, H., and Tilly, J.L. Oocyte formation by mitotically active germ cells purified from ovaries of reproductive-age women. Nat Med. 2012; 18: 413–421.

Yi, Y.C., Chen, M.J., Ho, J.Y., Guu, H.F., and Ho, E.S.Mitochondria transfer can enhance the murine embryo development.J Assist Reprod Genet.2007;24:445–449.

El Shourbagy, S.H., Spikings, E.C., Freitas, M., and St. John, J.C.Mitochondria directly influence fertilisation outcome in the pig.Reproduction.2006;131:233–245.

Hua, S., Zhang, Y., Li, X.C., Ma, L.B., Cao, J.W., Dai, J.P. et al.Effects of granulosa cell mitochondria transfer on the early development of bovine embryos in vitro.Cloning Stem Cells.2007;9:237–246.

Fakih, M.H.S.M., Szeptycki, J., dela Cruz, D.B., Lux, C., Verjee, S., Burgess, C.M., Cohn, G.M., and Casper, R.F. The AUGMENT treatment: physician reported outcomes of the initial global patient experience. JFIV Reprod Med Genet. 2015; 3: 154.

St John JC, Makanji Y, Johnson JL, Tsai TS, Lagondar S, Rodda F, Sun X, Pangestu M, Chen P, Temple-Smith P. The transgenerational effects of oocyte mitochondrial supplementation. Sci Rep. 2019 Apr 30;9(1):6694.

2) "There is no detail about the depth of the sequencing. Ion Torrent is not a reliable sequencing platform – it produces too many unreliable reads – you need to use a more reliable platform such as MiSeq. The levels of heteroplasmy need to be assessed on a range of tissues to determine if you see uniform similarities in transmission. At the moment, these data are not convincing as too few tissues are assessed.""Conclusions on mtDNA heteroplasmy are weak because results are based on detection of only one SNP difference between NZB/OlaHsd and B6CBAF1 strains. It is likely that mtDNA haplotypes of these two strains differ on more than 90 SNPs. Therefore, authors should monitor heteroplasmy on each of these positions. This should be done by whole mtDNA sequencing, which is required for conclusive heteroplasmy assays post MRT. "Please provide additional sequencing data to solidify this key point.

We agree that depth of sequencing should be included in the manuscript to show reliability of data. In fact, care was taken to ensure that each sample was sequenced at great depth (mean: ~60,000 reads) to ensure high accuracy and sensitivity of variant calling. We now included mean sequencing coverages (± SDs and ranges) of alleles at the variant site in footnotes of Supplementary file 1, Supplementary file 2 and Supplementary file 4. In addition, we prepared an excel file for reviewers with source data for these analyses (see Source data files), listing exact sequencing coverages at the variant site in all processed samples. These files, together with footnotes included in relevant tables, show that sequencing depth was consistently high and sufficient for variant calling throughout all samples tested.

We agree that it is essential to obtain accurate sequencing reads when determining variant ratios. We believe that the Ion Torrent is a highly reliable sequencing platform and well‑suited for variant analysis. It is routinely used for this purpose in research as well as clinical diagnostics and forensics (see referencesYang et al., 2018; Singh et al., 2013; Heeke et al., 2018; Churchill et al., 2018; below for recent examples). Furthermore, comparison of the Ion Torrent to other sequencers, including MiSeq, has not shown differences in performance (see referenceQuail et al., 2012). In our laboratory we use both platforms, the MiSeq and the Ion Torrent. For the present project we decided to use the Ion Torrent because it allows deep sequencing at much lower cost and much higher sample throughput than the MiSeq. Importantly, the Ion Torrent is coupled with a powerful server for direct processing of generated sequencing data, utilising software specifically developed for Ion Torrent raw data analysis. The verified Torrent Variant Caller plug‑in was used to determine variant ratios. Optimised, pre-set parameters were applied for variant calling. Some parameters were further customised to increase variant calling accuracy: (1)Settings for “Somatic” variants analysed at “High Stringency” were applied to enable low frequency variant detection at a minimal false‑positive call rate. (2)The read depth was down sampled to 20,000, rather than 2,000 reads as recommended, to increase accuracy of variant calls. (3)A “HotSpot Region” BED file, defining the exact genomic coordinate of the assessed nucleotide, in addition to a “Target Region” BED file, was applied to increase variant calling sensitivity at the assessed locus. Additionally, sequencing results of all samples were visualized in the integrative genomics viewer (IGV) to confirm variant calls and to ensure variant calls/low-level heteroplasmies were not ignored. Overall, we believe that our data is very accurate. Of note, in addition to the known variant site we also assessed all other nucleotides in sequences flanking the variant site. No sequencing artefacts/errors were detected in any position in the ~100bp amplicons in any of the samples.

We have edited the Materials and methods section to further clarify this point.

We agree that it is important to assess heteroplasmy levels in a range of adult mice tissues to determine mtDNA transmission and possible unequal distribution. For these reasons, we tested a total of 110organs and tissues of different energy needs (22tails, 22hearts,22livers, 22brains and 22kidneys). These were derived from 22animals (10female and 12male mice) from five consecutive generations (6xF1, 4xF2, 4xF3, 4xF4 and 4xF5, please see Supplementary file 4). Detected heteroplasmy levels were always comparable between organ/tissue types of individual mice and between mice of same generation. Moreover, heteroplasmy levels in organs/tissues of F1mice were comparable to the heteroplasmy levels detected in embryo biopsies. This data suggests that the types of organs/tissues and numbers of mice/generations analysed were sufficient to assess mtDNA transmission. Of note, published studies of similar type assessed similar numbers and organs/tissues (see referencesJenuth et al., 1997; Sharpley et al., 2012) to what we present here.

Accurate mtDNA heteroplasmy levels can be obtained from the analysis of a single SNP, since all polymorphic sites are inherited together. In fact, the analysis of a single polymorphic nucleotide is routinely used for the quantification of heteroplasmy levels and mtDNA carry-over rates (see referencesJenuth et al., 1997; Sharpley et al., 2012; Tachibana et al., 2009; Hyslop et al., 2016). The SNP at position m.3932 was used for mtDNA quantification in the present study. Differing B6CBAF1 and NZB alleles were initially confirmed by minisequencing. High accuracy and high sensitivity in heteroplasmy detection using the specified polymorphic nucleotide was confirmed using artificial mixtures of genomic DNA (see Supplementary file 1). Of note, the analysis of single nucleotides for mtDNA heteroplasmy quantification is a well‑established procedure in our laboratory and has been extensively validated on a range of samples (including human) for several other research projects as well clinical diagnostics. Therefore, we believe that the conclusions drawn in the present study are legitimate.

3) "All pups (F2) were born alive, respired normally and grew to adulthood without manifesting any physiological or behavioral alteration.” and “Gross necropsies of the parents and offspring were also performed during the 5 generations, with no pathological findings observed. Where are these data and how much depth was there to these analyses? This is critical given that a recent paper has shown that autologous mitochondrial supplementation can lead to a heart defect." Clarify to what depth mice were studied at autopsy.

The health, reproductive performance and welfare were monitored in all mice born from F1 until F5 resultant from the MST-derived lineage (in total, 239 mice). All mice analysed grew healthy and those selected for mating presented good fertility condition. Additionally, at weaning (21 days after birth), all pups showed normal appearance and appropriate size with no differences detected among gender, nor were any episodes of premature death reported. A gross necropsy of all parental mice (F1-F5) was performed, meaning that organs were observed macroscopically with no pathological findings observed, in terms of size, texture or morphological appearance, as detailed in subsection “Developmental potential of MST mice and mtDNA heteroplasmy fate” of the revised manuscript. In addition, some organs (tail, heart, brain, kidneys and liver) were harvested for analysis of mtDNA heteroplasmy, as stated in the manuscript.

4) "Authors used NZB strain as an example of a poor breeder strain. Strikingly, F1 females derived from MST that are still genetically NZO produced large litters (~8 pups) according to Supplementary file 1. It seems that authors show only one litter, which seems to be large even in unmanipulated NZO strain. It would be useful to show if F1 females maintain high productivity over time compared to NZO females (which should be also included in the table). If MST transfer improved NZO female's fecundity this could indicate that their breeding problems are mitochondria related and replacing mitochondria could improve the performance of this strain. Which would be of interest also from the clinical perspective. This technique could potentially reveal mitochondrial causes of infertility in patients which can be hereditary in the female lineage Women treated for ooplasm deficiency could pass this to their daughters." Please comment on the reviewer's observation.

We thank the reviewer for this interesting observation. Our data indicate that over the five generations the MST-derived mice were crossed and followed, the litter size was large and stable (average of 7.4 pups, see Supplementary file 3). Based on the literature and information provided by the commercial provider (Invigo), poor reproductive performance is expected for NZB/OlaHsd mice, with litter sizes of 3.8 at weaning and colony output 0.5 young/female/week (Festing, 1976; Fernandes et al., 1973; Hansen et al., 1973). Unfortunately, we did not follow the NZB controls after F1, as the main aim of our study was to evaluate the health, welfare and reproductive performance of the mice resultant from MST. We could hypothesize based on this observation that the MST technique can potentially reveal mitochondrial or cytoplasmic causes of infertility, but we do not have a direct control group to support this finding. Our group is currently investigating this question as part of another study that is being carried out.

We have added this to the Discussion of the revised manuscript.

[Editors' note: further revisions were suggested prior to acceptance, as described below.]

Reviewers remain concerned about the strength of the evidence to show that the animals were indeed normal. Were any organs preserved for histological examination? If so, it would be reasonable to examine at least a subset of these. The reviewers feel that given the impetus to take this technology to the clinic, this information from your rather extensive preclinical study is of considerable importance. Any additional data related to this point would be most valuable.Reviewer #1:Whilst I am not convinced by the explanation for the use of Ion Torrent, it is not my primary concern.I am primarily concerned about the lack of detail related to the health and well-being of the offspring. The subsection “Developmental potential of MST mice and mtDNA heteroplasmy fate” does not provide sufficient information to determine if the animals were healthy or not and no data are presented. There is also no detail in the Materials and methods section. Macroscopic and microscopic investigations need to be performed. Currently, there is a drive to push this form of assisted reproductive technology into clinical practice in various countries and the real question relates to the health and well-being of the offspring. A number of studies have offered accounts of the fate of mtDNA that is carried over. Your study requires some novelty and macroscopic and microscopic investigations would provide this.I am not prepared to support the publication of a manuscript related to this form of assisted reproductive technology that does not deal with the crucial question about the health and well-being of the offspring. Three lines of text is simply not sufficient.

Following this reviewer’s suggestions we have performed additional MST experiments both using both NZB and B6 strain oocytes. Blastocysts were transferred and 12 adult animals were examined in detail through histopathological analyses. For this set of experiments, necropsies and histological examination of the 12 animals (4 from resultant from ICSI in B6 strain, 3 from reciprocal MST in B6 and 5 NZB-sp/B6-cyt MST assays) was done blindly by the pathologist (see attached report where G1 = ICSI-B6, G2 = B6-sp/B6-cyt MST, G3 = NZB-sp/B6-cyt MST). We have also included representative findings in a new figure: Figure 4—figure supplement 1).

We have updated accordingly the Results section:

“Gross necropsies of the parents and offspring were performed during the 5 generations, with no pathological findings observed. In the 239 mice analysed, all organs showed a normal size, texture and morphological appearance. Additionally, histopathological examinations were performed in organs including heart, kidney, liver and brain, in the F1 mice generated for this purpose by MST B6-sp/B6-cyt (n=3), MST NZB-sp/B6-cyt (n=5) and control B6 (n=4) groups. Except for a pericardium focal inflammation in one animal of the B6 control group, neither of the animals showed any lesions or visible abnormalities (Figure 4—figure supplement 1). Taken together, these results support the notion that MST can efficiently produce healthy, fertile and viable offspring.”

And the Materials and methods section (Histological analysis):

“For histological evaluation, tissue samples from heart, liver, kidney and brain of 4 ICSI-B6 control, 3 B6-sp/B6-cyt MST and 5 NZB-sp/B6-cyt MST mice were collected at 6 weeks of age. Mice were first perfused with PBS and then with 5% formaldehyde solution. Subsequently, tissues were fixed overnight at 4°C in 5% formaldehyde and embedded in paraffin wax, sliced in 4-μm sections and stained with hematoxylin and eosin staining (H&E). The atrium, valves and myocardium of heart; both kidneys (through longitudinal and transverse sections), liver and gall bladder, forebrain, midbrain and hindbrain were evaluated. Histological analysis was carried out blindly using mouse identification codes for group assignment that were unknown to the evaluator.”

As stated by the pathologist in the report attached, macroscopic observations revealed that all animals were well nourished, well groomed and active; with no dermal lesions, nasal, ocular or genital discharges seen. Thoracic and abdominal viscera did not show abnormalities either. Additionally, histological analysis tissues performed in heart, kidney, liver and brain sections showed no apparent lesions in any of the organs evaluated in mice derived from *NZB sp-/B6-cyt* MST and *B6-sp/B6-cyt* MST embryos. Neither of the animals evaluated presented lesions of significance, except for one mouse derived from B6-ICSI control group that showed a pericardium inflammation.

In addition, we have also performed an extensive comparison between Ion PGM and Illumina MiSeq sequencing platforms using up to five different allele variants in tissues with known different heteroplasmy levels, please see detailed answer to reviewer #3 below.

Reviewer #2:Authors provided additional information requested by reviewers such as references, added text and supplementary data. Other comments were addressed in the rebuttal by providing references supporting their argument but without adding additional data (although cited papers did not always agree – see point 1).I am satisfied with most of the issues addressed but would like to hear from other reviewers if their comments were sufficiently addressed.1) Addressing comment 2 that 5 tissues are insufficient to determine transmission of heteroplasmy authors cited two papers (Sharpley et al., 2012; Jenuth et al., 1997) saying that they have tested "a similar number" of tissues when in fact they used twice as many (10 or 9).

We are glad that the information provided in the previous revision mostly satisfied reviewer #2. We have included histological evidence that supports the notion that the mice generated by spindle transfer are normal and provided additional data that confirms the reliability of the Ion PGM platform used to determine heteroplasmy in the different organs of the mice of the different generations.

Reviewer #3:I believe that the revision did not address the main concern expressed by the reviewers that whole mtDNA sequencing (MiSeq) is required to validate conclusions on heteroplasmy. It is standard now to validate sequencing by two independent approaches. The author's arguments in the rebuttal are not convincing.

As indicated in the previous revision of our manuscript, comparison of the Ion Torrent to other sequencers, including MiSeq, had not shown differences in performance (see reference Quail et al., 2012).

Nevertheless, we decided to address the reviewers concern by first doing a direct comparison of the performance and accuracy of the IonTorrent PGM machine and Illumina’s MiSeq; and secondly by not only assaying 1 SNP from one mtDNA amplicon, but for a total of 3 amplicons, analyzing a total of 5 mitochondrial SNPs (B6CBAF1/NZB: m.3932A/G, m.2798C/T; m.2814T/C; m.3194T/C; m.3260A/G):

We describe this in detail in the Materials and methods section:

“To ensure both the validity of assessment of a single SNP for mtDNA carryover analysis and the accuracy of the utilized sequencing platform (Ion PGM); a second set of experiments was performed, which included analysis of four additional SNPs (B6CBAF1>NZB: m.2798C>T; m.2814T>C; m.3194T>C; m.3260A>G) utilizing Illumina’s MiSeq System sequencing platform. […] Furthermore, gDNA samples from organs of three individual mice were sequenced and results compared to those obtained by PGM sequencing.”

We have also updated the Results section accordingly :

“To verify validity of mtDNA carryover assessment by analysis of a single SNP and to ensure reliability of the utilised sequencing platform, four additional SNPs (B6CBAF1/NZB: m.2798C/T; m.2814T/C; m.3194T/C; m.3260A/G) were analysed on a different sequencer (Illumina’s MiSeq). The presence of different alleles was also confirmed by minisequencing”

(see Materials and methods and Supplementary file 2 for further details; and Figure 3—figure supplement 2). And in the Results section:

“These quantifications based on a single SNP in an Ion PGM sequencer were corroborated by using an additional sequencing platform (Illumina’s MiSeq) and 5 SNPs, as described above. Artificially constructed samples, composed of gDNA from both mouse strains mixed in different ratios, and gDNA from 5 organs of selected adult mice from F1-3 generations were analysed (Figure 3—figure supplement 2, Figure 4—figure supplement 2, Supplementary files 2 and 6). These results suggest that low levels of mtDNA heteroplasmy resultant from MST typicallly result in a homoplasmic state in offspring within a few generations, without reversion (Supplementary files 5 and 6).”

These results are summarized in new Supplementary files 2 and 6 (with the corresponding source data in 2 data files). The direct comparison of measurements of 5 organ samples from MST mice from 3 generations between the 2 sequencing platforms is depicted in Figure 4—figure supplement 2.

Overall, our analyses show that there are no significant differences between the heteroplasmy levels regardless of whether we examine 1 mtDNA SNP or more; or the sequencing platform utilised. These new validations confirm that Ion PGM is reliable and well-suited for this type of variant analysis. Of note, the heteroplasmy levels of the offspring generated by MST is almost null from generation F3 (see Figure 4 and Figure 4—figure supplement 2 and Supplementary files 5 and 6), suggesting that the low mtDNA heteroplasmic levels result in homosplamic state in the offspring within a few generations and without reversion. Therefore, we believe that the conclusions drawn in the present study are legitimate.

[Editors' note: further revisions were suggested prior to acceptance, as described below.]

Reviewer #1:I am pleased to see that the authors have taken on extra work to seek to validate their claims regarding the safety of MST.I note that the authors have undertaken the histopathology analysis. However, I am surprised to see that they have restricted themselves to four organs. Likewise, as argued by one of the other reviewers, they do not carry out mtDNA analysis on all tissues. Therefore, the arguments they present are only partially valid. The reason for this is that mtDNA, and specifically mtDNA mutations (as in disease) or variants (in non-pathological situations), do not segregate neutrally, or evenly, amongst all the tissues/organs but rather in a random manner. Therefore, without having analysis of all tissues and organs (or at least 10 as agreed by another reviewer), the outcomes are only indicative of what has been analysed. I, further, note that muscle has not been analysed and myopathies are associated with mtDNA disease and, in the disease state, muscle often carries high loads of mutant mtDNA, which affects OXPHOS function (mtDNA encodes for key genes of the electron transfer chain that performs OXPHOS).In conclusion, I think this is an important study but it is being held back though either reluctance or a lack of understanding of mtDNA genetics and, specifically, transmission and segregation of the mitochondrial genome. Either way, this results in the conclusions being inconclusive.

In the previous version of our revised manuscript we had included histopathological examinations that had been performed in vital organs, including heart, kidney, liver and brain. We have now extended these examinations to additional tissues from the same animals, as suggested. We included the analysis in both skeletal tibial and quadriceps muscle and urinary bladder for smooth muscle. Additionally, we have also processed organs from the reproductive tract and accessory glands in both males (testis, epididymis, seminal vesicles, prostate, coagulating glands, ampullary glands and bulbourethral glands) and females (ovaries, oviducts, uterine horns). Taken together we have examined over 10 organs/tissues from each animal. None of the animals showed any lesions or visible abnormalities. Representative images from additional stainings are now included in Figure 4—figure supplements 1 and 2.

On the other hand, we have tried to tone down the claims that the MST animals were “healthy” and emphasized in the revised Discussion that the conclusions regarding the animals health and heteroplasmy levels are drawn only from the organs we assessed.

References:

Churchill, J. D., Stoljarova, M., King, J. L. & Budowle, B. Massively parallel sequencing-enabled mixture analysis of mitochondrial DNA samples. Int. J. Legal Med. 132, 1263–1272 (2018).

Festing MF. Phenotypic variability of inbred and outbred mice. Nature. 1976 Sep 16;263(5574):230-2.

Heeke, S. et al. Use of the Ion PGM and the GeneReader NGS Systems in Daily Routine Practice for Advanced Lung Adenocarcinoma Patients: A Practical Point of View Reporting a Comparative Study and Assessment of 90 Patients. Cancers (Basel). 10, 88 (2018).

Hyslop, L. A. et al. Towards clinical application of pronuclear transfer to prevent mitochondrial DNA disease. Nature 534, 383–386 (2016).

Jenuth, J. P., Peterson, A. C. & Shoubridge, E. A. Tissue-specific selection for different mtDNA genotypes in heteroplasmic mice. Nat. Genet. 16, 93–95 (1997).

Quail, M. et al. A tale of three next generation sequencing platforms: comparison of Ion torrent, pacific biosciences and illumina MiSeq sequencers. BMC Genomics 13, 341 (2012).

Sharpley, M. S. et al. Heteroplasmy of mouse mtDNA is genetically unstable and results in altered behavior and cognition. Cell 151, 333–343 (2012).

Singh, R. R. et al. Clinical Validation of a Next-Generation Sequencing Screen for Mutational Hotspots in 46 Cancer-Related Genes. J. Mol. Diagnostics 15, 607–622 (2013).

Tachibana, M. et al. Mitochondrial gene replacement in primate offspring and embryonic stem cells. Nature 461, 367–372 (2009).

Yang, D. et al. An SNP panel for the analysis of paternally inherited alleles in maternal plasma using ion Torrent PGM. Int. J. Legal Med. 132, 343–352 (2018).